# Study on Mechanical Properties and Durability of Alkali-Activated Silicomanganese Slag Concrete (AASSC)

**Baifu Luo [1,*], Dong Wang [1] and Elchalakani Mohamed [2]**

1   College of Civil Engineering and Mechanics, Xiang Tan University, Yang Gu Tang Street, Xiangtan 411105, China
2   School of Engineering, Civil, Environmental and Mining Engineering, The University of Western Australia (M051), 35 Stirling Highway, Perth 6009, Australia
*   Correspondence: luobaifu@xtu.edu.cn

**Abstract:** Alkali-activated materials are produced by chemically polymerizing the aluminosilicate materials using alkaline activators, which can effectively lower the greenhouse-gas emissions (approximately 73%) released by ordinary Portland cement (OPC). Silicomanganese slag is a large solid waste discharged from the ferroalloy industry in China that can pollute the environment and occupy resources. In this paper, the slag in alkali-activated material was replaced with silicomanganese slag to address the disposal of silicomangaese slag. The flowability, setting times, compressive and flexural strengths, micro-structure and freeze-thaw resistance of alkali-activated silicomanganese slag concrete (AASSC) with varied substitution ratios, volume fractions of steel fibers and alkali-activated modulus ($M_S$) were exploited. As a results the compressive strengths at 56 days of AASSC with a 10% substitution ratio of silicomanganese slag reached over 80 MPa and over 132 MPa with the 3% steel fiber dosage. AASSC still reached 91 MPa and 45 MPa with 60% and 100% substitution ratios by cooperating 2% steel fibers, respectively. When the freeze-thaw cycle number reached 300, the compressive strengths of AASSC with the replacement ratios of 10%, 60% and 100% were 84%, 74% and 51% of their original values by cooperating 2% steel fibers, respectively; AASSC with the numerous substitutions of 60% and 100% were destroyed at 600 and 300 freeze-thaw cycles, respectively. AASSC with a 10% substitution ratio and 2% steel fiber content is suitable for excellent performance, and a 60% substitution ratio can also be applied to construction for the massive utilization of silicomanganese slag.

**Keywords:** mechanical properties; freeze-thaw resistance; micro-structure; alkali-activated; silicomanganese slag

## 1. Introduction

With further improvement and development in technology, silicomanganese alloys have been widely used in the steel industry. According to International Manganese Institute statistics, around 16 million tons of SiMn alloy was produced in 2018 worldwide, which increased by 16% from 2017, and production also continuously increased in 2019, with China sharing the majority [1]. It is about 1.3 tons of silicomanganese slag generated per ton of alloy production [2]. With the rapid development of silicon manganese alloy, the main metals used in China are silicon manganese alloy and low-carbon ferromanganese alloy, followed by medium-low ferromanganese alloy [3]. Silicomanganese slag is made of a variety of raw materials such as manganese ore, lime, silica, etc. The chemical components of silicon manganese slag mainly include silicon dioxide and calcium oxide, and the sum of both exceeds 55% with the silicomanganese slag having high potential hydration activity [4]. For the granular silicomanganese slag, the mineral compositions are mainly vitreous, have high activity and contain fewer minerals such as $AlSiO_5$ [5]. Silicomanganese slag can be better ground than other metal slag due to its internal composition. After testing the

density and specific surface area, researchers also found that the specific surface area of granular silicon manganese slag was larger than that of other massive silicomanganese slags [6]. It was found that silicon manganese slag could complement cement and be used as cementitious materials in concrete. Silicomanganese slag has a good activity that could affect the performance of concrete [7].

Due to the rapid development of the steel industry, a large amount of metallurgical slag needs to be recycled. Explorations have been made into the recycling and utilization of metallurgical slag. Metallurgical slag has largely used in cement production, concrete admixture and road materials, but metallurgical slag has the characteristics of unstable internal chemical composition and low activity, which can reduce its practical application. To improve the activity and utilization of metallurgical slag, physical activation, chemical activation and thermal activation have mainly employed in previous literature.

(1)    Physical activation

The application of mechanochemistry technology in environmental protection is a research hotspot [8]. The common form of physical activation is mechanical grinding. Through mechanical grinding, there is a large amount of mechanical energy in the metallurgical slag, and the surface has a high activity. Physical activation not only cause physical and chemical changes in its internal structure, but develop also obvious micro-cracks on the surface of the material. At the same time, crystal defects and lattice distortion occur in the structure, which accelerates the ability of other substances to participate in the reaction [9].

(2)    Chemical activation

The activators can stimulate and improve the intrinsic activity of metallurgical slag for hydration. Common activators include acid–base activators, seed excitation [10], sulfate activators [11], chloride activators [12], etc., which can promote the disintegration of metallurgical slag and generate stable hydration products.

(3)    Thermal activation

The common methods of thermal activation are high-temperature calcination and quenching [13], and the use of high-temperatures with the addition of structural adjusters [14].

The popular research concentrating on silicomanganese slag has been applied to the substitution of cementitious material in concrete. He et al. [15] investigated the feasibility of preparing belite–calcium sulphoaluminate cement by using a synergistic reaction of electrolytic manganese residue and barium slag, as well as limestone, and bauxite. The results showed that the compressive strengths of samples at 3, 7, and 28 days were 37.9, 47.3, and 60.0 MPa, respectively, which were better than those of OPC425. Wang et al. [16] utilized cement, sand, crushed stone, water reducer and silicomanganese slag powder to prepare C60 (compressive strength $f_{cu}$ equals 60 MPa) concrete. The mechanical and durability performance of concrete was enhanced by substituting 10–20% of cement with ultra-fine silicomanganese ash, because the silicon manganese ash can react with the activator gypsum or lime and also fill the gaps between cement and particles. It was preliminarily estimated that each cubic meter of concrete could save 200 kg of cement. Therefore, the application of silicomanganese slag in high-strength concrete can achieve economic and mechanical effects. Srikavya et al. [17] replaced natural fine aggregates and coarse aggregates with silicomanganese slag by 0%, 5%, 10%, and 15%, respectively. The result indicated that 15% substitution ratios to natural fine aggregates and coarse aggregates could reduce a certain amount of aggregates in concrete with a limited strength reduction. However, these methods of substituting cement in concrete had a limited substitution ratio, and high-strength concrete-containing silicomanganese slag required the use of ultra-fine particles and had a limited ratio of 20%, which restricted its practical utilization.

Alkali-activated materials are seen as the alternative materials to cement-based concrete in recent years based on their low carbon emission and the use of amorphous aluminosilicate-based powder, it is activated by alkalis such as granulated blast furnace slag (GBFS) and fly ash. Zhang et al. [18] studied the durability of alkali-activated concrete

and found out that the durability of alkali-activated concrete was better than cement-based concrete under the same strength level. A research review [19] indicated that the early and ultimate mechanical strengths of alkali-activated cementitious systems are better than that of the traditional cement-based binder. Ambily et al. [20] reported that UHPGC with the highest compressive strength of 175 MPa was successfully produced by mixing steel fibers under curing at ambient temperature. You et al. [21] also reported that ultra-high-performance alkali-activated concrete can be applied to structures with a lightweight. A research review [22] studied the properties of ultra-high-performance alkali-activated concrete by considering variables such as the steel fibers and activators, which discovered that the ultra-high performance alkali-activated concrete is an eco-friendly material with great mechanical strength and durability. However, the potential problem of alkali-activated materials is their costs, which are higher than those of ordinary cement-based concretes.

In the previous literature, Su et al. [23] utilized silicomanganese (SiMn) slag (ultra-fine powder, no more than 3% above 80 μm) as the cementitious material, activated by sodium hydroxide. The results indicated that SiMn slag can be applied to construction with high strength. Huang et al. [24] used three common activators to study the mechanical properties of alkali-activated materials with the addition of granular silica manganese slag and analyzed the effects of anhydrous sodium sulfate, sodium hydroxide and water glass on the performance of cementitious materials. The strength of the materials with sodium hydroxide addition increased rapidly in the early stage and slowly in the later stage. It was appropriate to use 4% sodium hydroxide to excite the cementitious materials, which could gain a better effect with water glass addition. Through comparative analysis of the three materials, it was found that sodium hydroxide is the best among all materials when the content of the activator is lower than 3%. Nath et al. [25] added granular silicon manganese slag to fly-ash-based alkali-activated cementitious material to replace a part of the fly ash. When the silicon manganese slag content increased, the reaction rate of the material accelerated. Navarro et al. [26] studied the varied activators on the alkali-activated SiMn slag pastes. The results showed that the main reaction product formed in the alkaline activation of SiMn slag is C-S-H with a low Ca/Si ratio, and it is higher for pastes activated with NaOH compared to those with water glass. The method of using activators for silicomanganese slag generally requires massive a ground effort for high-strength. The way to substitute the cementitious material (fly ash) uses a small substitution ratio. So, it is worthwhile to research a new way to improve the utilizable amount of silicomanganese slag with excellent strength to meet engineering needs.

This paper used silicomanganese slag powder to replace the grounded blast furnace slag to generate alkali-activated silicomanganese slag concrete (AASSC) with excellent performance. The main cementitious materials of slag or fly ash in alkali-activated concrete, generally have a greater percentage in alkali-activated concrete in comparison to the cement in cement-based concrete, which implies that silicomanganese slag can be utilized more in alkali-activated concrete than in cement-based concrete. Compared to silicomanganese slag in alkali-activated concrete in a previous study, the method in this study can activate the grounded blast furnace slag and silicomanganese slag both by sodium hydroxide, and silicomanganese slag powder can also act as a filler to fill the micro-voids inside cementitious materials, resulting in improved strength and durability. To systematically analyze the AASSC, the mechanical properties, freeze-thaw resistance and micro-structures of AASSC were studied by considering variables such as the substitution ratio of silicomanganese slag, alkali-activated modulus ($M_S$) and volume fraction of steel fiber. The list of abbreviations is shown in Table 1.

## 2. Raw Materials and Experimental Details

### 2.1. Raw Materials

Raw materials used in this study were fly ash (FA), silica fume (SF), grounded blast furnace slag (GBFS), fine sand and sand, water (W), steel fibers, polycarboxylic acid superplasticizer (SP), NaOH and water glass (WG). The chemical and physical properties

of the main materials are shown in Tables 2–6 and Figures 1 and 2. The silicomanganese slag used in the test was provided by an alloy factory in Anyang, Henan Province. The treatment of silicomanganese slag encompassed grinding for 3 h. The silicomanganese slag mainlycontains Si, Ca, Mg, Al, Mg and other elements, and has certain hydraulic and volcanic properties.

**Table 1.** List of abbreviations.

| Acronym | Abbreviations |
|---|---|
| FA | Fly ash |
| SF | Silica fume |
| GBFS | Grounded blast furnace slag |
| AAMs | Alkali-activated materials |
| AASSC | Alkali-activated silicomanganese slag concrete |
| UHPGC | Ultra-high performance alkali-activated concrete |
| LOI | Loss of ignition |
| WG | Water glass |
| SP | Polycarboxylic acid superplasticizer |
| XRD | X-ray diffractometer |
| SEM | Scanning Electron Microscope |
| C-S-H | hydrated calcium silicate |
| $M_S$ | Alkali-activated modulus |

**Table 2.** Chemical composition of raw materials (%).

| Materials | $SiO_2$ | $Al_2O_3$ | $K_2O$ | $TiO_2$ | $SO_3$ | $Na_2O$ | MgO | CaO | $Fe_2O_3$ | LOI |
|---|---|---|---|---|---|---|---|---|---|---|
| FA | 53.36 | 29.09 | - | - | | - | 0.81 | 2.27 | 3.87 | 2.48 |
| SF | 95.26 | - | 0.98 | - | - | 0.98 | 0.29 | 1.93 | 0.57 | 2.48 |
| GBFS | 34.21 | 14.15 | 1.95 | - | 0.12 | - | 10.32 | 39.11 | 0.82 | 0.7 |
| Silicomanganese slag | 36.27 | 7.82 | - | 0.67 | 0.74 | 0.28 | 11.01 | 26.49 | 0.79 | 3.98 |

**Table 3.** Physical properties of GBFS and SF.

| Material | Specific Surface Area ($m^2$/kg) | Specific Density (kg/$m^3$) | Mass Coefficient (Kkc) | 28 Days Activity Coefficient (%) |
|---|---|---|---|---|
| GBFS | 519 | 2980 | 1.697 | 110 |
| SF | 2500 | 2700 | - | 121 |

**Table 4.** Particle size of GBFS and SF.

| Particle Size (μm) | <2 | <4 | <10 | <16 | <20 | <32 | <64 | <80 | <100 |
|---|---|---|---|---|---|---|---|---|---|
| GBFS | 4.1 | 12.78 | 36.3 | 41.8 | 49.5 | 65.9 | 87.2 | 94.2 | 100.0 |
| SF | 98.21 | 100.0 | 100.0 | 100.0 | 100.0 | 100.0 | 100.0 | 100.0 | 100.0 |

**Table 5.** Properties of fine sand and sand.

| Material | Particle Size (mm) | Bulk Density (kg/$m^3$) | $SiO_2$ (%) | $Fe_2O_3$ | Water Content (%) |
|---|---|---|---|---|---|
| Fine sand | 0.08–0.15 | 1600 | 90.5–92.7 | 0.03–0.10 | 1.5 |
| Sand | 0.178–0.425 | 1450 | 90.5–92.7 | 0.03–0.10 | 1.5 |

**Table 6.** Properties of steel fiber.

| Shape | Length (mm) | Diameter (mm) | Density (g/cm$^3$) | Roughness | Tensile Strength (MPa) | Elongation (%) |
|---|---|---|---|---|---|---|
| Straight | 6 | 0.16 | 2.56 | smooth | 2500 | 3.5 |

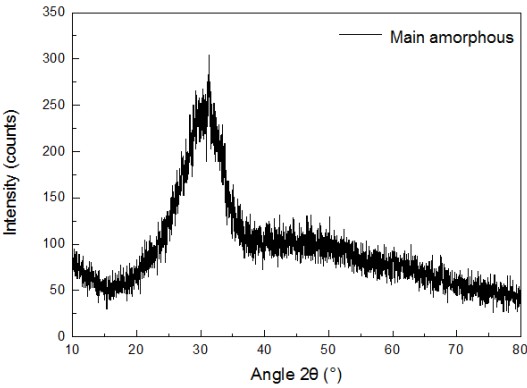

**Figure 1.** XRD patterns of GBFS.

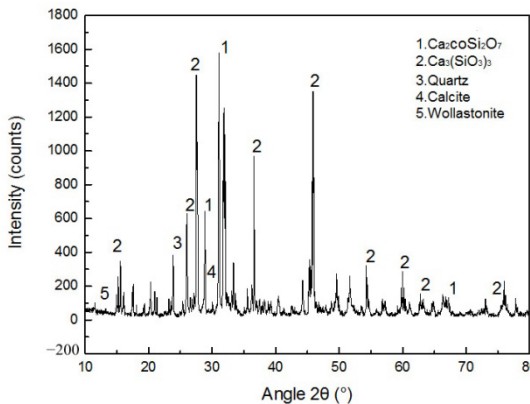

**Figure 2.** XRD patterns of silicomanganese slag.

As seen in Figure 1, the mineral composition of the GBFS is amorphous with high activity, which means that GBFS has high pozzolanic effects and can fill the inner voids in the matrix to improve the density of the alkali-activated concrete. As seen in Figure 2, the main mineral components of the silicomanganese slag are calcium silicate, cobalt chromite, quartz, calcite and wollastonite. The activity of the silicomanganese slag can be improved by grinding.

### 2.2. Proportioning and Mixing Details of Study

As shown in Table 7, the water-to-binder ratio used in this research was 0.32, the sand-to-binder ratio was 1.0, and the alkali content (calculated by $Na_2O$) was 7%, which was derived from previous research [27] and revised on our own raw materials. The variables such as the replacement percentage of GBFS, alkali-activated modulus ($M_S$) and steel fiber content of the AASSC were studied. Group S10–S30 were the factors related to the ratios of GBFS replaced with silicon manganese slag (10–100%), S10-M1 to S10-M2 were factor related to modulus $M_S$ (1.0, 1.5 and 2.0), S10-SF1 to S10-SF3were factors related to volume ratios of the steel fiber and the sample (1%, 2% and 3%).

**Table 7.** Mix proportion of AASSC ($kg/m^3$).

| No. | GBFS | FA | SF | Silicon Manganese Slag | Fine Sand | Sand | Steel Fiber (%) | SP | W | NaOH | WG |
|---|---|---|---|---|---|---|---|---|---|---|---|
| S0 | 688 | 172 | 45 | 0 | 271.5 | 633.5 | 0 | 13.8 | 36.2 | 36.2 | 416.3 |
| S10 | 619.2 | 172 | 45 | 68.8 | 271.5 | 633.5 | 0 | 13.8 | 36.2 | 36.2 | 416.3 |
| S20 | 550.4 | 172 | 45 | 137.6 | 271.5 | 633.5 | 0 | 13.8 | 36.2 | 36.2 | 416.3 |
| S30 | 481.6 | 172 | 45 | 206.4 | 271.5 | 633.5 | 0 | 13.8 | 36.2 | 36.2 | 416.3 |
| S10-M1 | 619.2 | 172 | 45 | 68.8 | 271.5 | 633.5 | 0 | 13.8 | 162.9 | 54.3 | 208.2 |
| S10-M1.5 | 619.2 | 172 | 45 | 68.8 | 271.5 | 633.5 | 0 | 13.8 | 97 | 45 | 314 |
| S10-M2 | 619.2 | 172 | 45 | 68.8 | 271.5 | 633.5 | 0 | 13.8 | 36.2 | 36.2 | 416.3 |
| S10-SF1 | 619.2 | 172 | 45 | 68.8 | 271.5 | 633.5 | 1 | 13.8 | 36.2 | 36.2 | 416.3 |
| S10-SF2 | 619.2 | 172 | 45 | 68.8 | 271.5 | 633.5 | 2 | 13.8 | 36.2 | 36.2 | 416.3 |
| S10-SF3 | 619.2 | 172 | 45 | 68.8 | 271.5 | 633.5 | 3 | 13.8 | 36.2 | 36.2 | 416.3 |
| S60-SF2 | 275.2 | 172 | 45 | 412.8 | 271.5 | 633.5 | 2 | 13.8 | 36.2 | 36.2 | 416.3 |
| S100-SF2 | 0 | 172 | 45 | 688 | 271.5 | 633.5 | 2 | 13.8 | 36.2 | 36.2 | 416.3 |

The letter 'S' represents the substitution ratio of silicomanganese slag, and the letter 'M' represents the modulus $M_S$. For example, S10–M1.5 indicates that the GBFS is replaced with a 10% substitution ratio of silicomanganese slag, and the modulus is 1.5. The letter 'SF' represents the volume ratio of steel fiber. For example, S10–SF1 indicates that the GBFS is replaced with 10% silicomanganese slag and the content of steel fiber is 1% of the sample volume.

Before the experiment, the alkali activator was prepared by mixing water, NaOH, and water glass for 24 h in advance. The preparation of samples was as follows. (1) Firstly, we mixed slag, fly ash, silica fume, and silicomanganese slag for 2 min at a low speed; (2) then, we added quartz sand for 1 min at a low speed; (3) finally, we slowly added alkali activator and mixed for 3 min, and the steel fibers were slowly added into the mixture for another 2 min. The fluidity test was conducted immediately after the mixing process with the conical mold, then mixtures were cast into the mold. A previous study [28] showed that 24 h of steam-curing at 80 °C could make alkali-activated slag concrete reach the strength development plateau, a method which was also applied in another study [29]. Therefore, a duration of 24 h steam-curing at 90 °C was applied in this study before the standard curing.

*2.3. Experimental Methods*

2.3.1. Fluidity and Setting Times

The fluidity tests of samples were conducted at room temperature, $20 \pm 2$ °C, and a humidity of more than 50%, and three specimens were tested for each mixture, according to the standard 'Test method for fluidity of cement mortar' (GB/T2419-2016) [30]. As seen in Figure 3, a Vicat instrument was used to measure the initial and final setting times of the samples. The tested method was as follows: (1) the inner sides of the mold were first coated with a thin layer of oil, and the round mold was placed on the glass plate; (2) we poured the slurry into the round mold, put it into the curing box, and started to record the setting time; (3) we recorded the samples as set when the pointer sank and the test needle could fall into the slurry without seeing a ring mark.

2.3.2. Mechanical Properties

The compressive strength tests of samples were prepared by pouring the slurry into the cubic mold of 40 mm, and bending strength tests were prepared by a $40 \times 40 \times 160$ mm prism mold according to a previous study [31]. All samples were covered with plastic film and cured by steam for 1 day after demolding, and finally placed in a standard curing room at a temperature of $20 \pm 5$ °C with humidity of more than 95%, based on standard DL/T5144-2001 [32]. The partial specimens after steam curing are shown in Figure 4a. Three specimens were tested for each mixture at the desired age for mechanical and flexural strengths (third-point bending as shown in Figure 4b) based on the Chinese standard GB/T 17671-2020 [33].

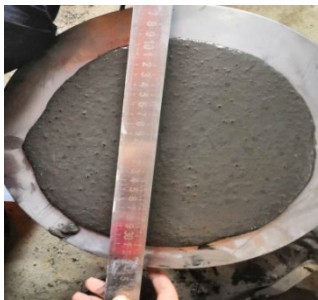

**Figure 3.** Liquidity test.

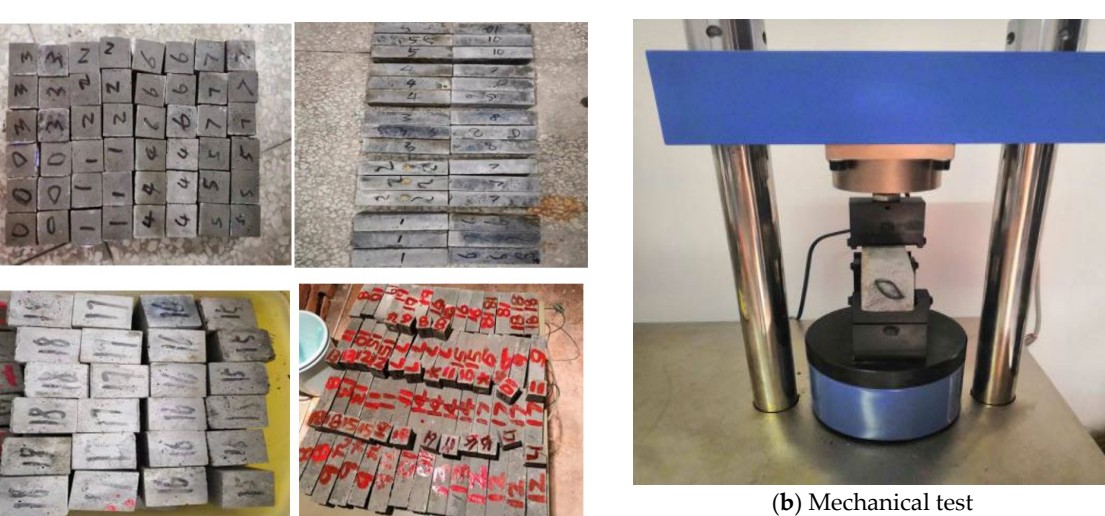

(**a**) Partial specimens

(**b**) Mechanical test

**Figure 4.** Samples and test details.

2.3.3. Freeze-Thaw Cycle Test

Three $40 \times 40 \times 40$ mm specimens from each mixture were applied for the freeze-thaw cycle test based on the Chinese standard GB/T50082-2009 [34]. The cycle time was 4.5 h, the temperature was set at $(-18 \pm 2)\sim(5 \pm 2)$ °C, and the number of freeze-thaw cycles was 300. Three specimens were prepared for each mixture and firstly immersed in distilled water for more than 0.5 h. After the immersion, we dried and weighed the samples accurately before the tests. The details are shown in Figure 5.

After n freeze-thaw cycles, the mass loss rates of the samples were calculated according to Formula (1):

$$W_{\mathrm{n}} = \frac{G_0 - G_{\mathrm{n}}}{G_0} \times 100\% \tag{1}$$

where $W_{\mathrm{n}}$ represents the weight loss rate of the sample after n times of freezing and thawing (%), $G_0$ represents the weight of the sample before the freeze-thaw test (g), $G_{\mathrm{n}}$ represents the weight of the sample after freezing and thawing (g).

After n freeze-thaw cycles, the compressive loss rates of the samples were calculated according to Formula (2):

$$Q_{\mathrm{n}} = \frac{P_0 - P_{\mathrm{n}}}{P_0} \times 100\% \tag{2}$$

where $Q_{\mathrm{n}}$ represents the loss rate of compressive strength of the sample after n freeze-thaw tests (%), $P_0$ represents the compressive strength of the sample before the test (MPa), $P_{\mathrm{n}}$ represents the compressive strength of the sample after the test (MPa).

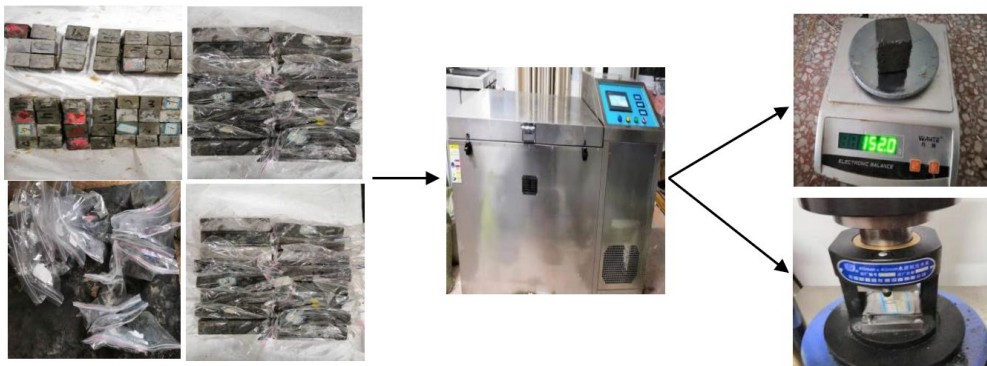

**Figure 5.** Apparatus and process of freeze-thaw cycle test.

### 2.3.4. Micro-Structure Analysis

X-ray diffractometer (XRD) tests were prepared by observation on sliced samples (2.5–5 mm). Sliced samples were firstly immersed in absolute alcohol for 24 h to stop the hydration, then ground into powders with a particle size of less than 0.08 mm; secondly, 20 g powders were took into a drying oven at $(60 \pm 2)$ °C for 48 h; finally, powders were tested at the Changsha Research Institute of Mining and Metallurgy. The working voltage used in the test was 32 kV. After the sliced samples were washed with ethanol and dried, the surface was sprayed with platinum to ensure conductivity. The Jim-6490lv scanning electron microscope was used in this test at Changsha Research Institute of mining and metallurgy.

## 3. Experimental Results

### 3.1. Working Performances of Samples

#### 3.1.1. Fluidity of Slurry

It can be seen in Table 8 that the method of replacing GBFS with silicon manganese slag had improved the fluidity of AASSC. The slurry with a larger replacement ratio tended to have a greater fluidity. The reason may be that the alkali-activated concrete with less slag has fewer silicon aluminum products, and silicomanganese slag with low activity slows down the early reaction. With the increase of modulus, it was found that the fluidity of slurry with a modulus 2.0 was 7.9% and 2.2% higher than slurries with modulus 1.0 and modulus 1.5, respectively. This is because the mixture with a higher modulus has a better fluidizing effect on extra silicate content [35] and a higher reduction in solid concentration [36]. When the content of steel fiber increased from 1% to 3%, the fluidity of concrete changed from 225 mm to 200 mm. The reason can be attributed to the fact that steel fibers are disorderly distributed in the mixture, which increases the friction and shearing force of the mixture [37]. In addition, steel fibers adsorb a certain amount of moisture on their surface, which reduces the free water content of the material, resulting in a decrease in the fluidity of the mixture [38].

**Table 8.** Fluidities of slurries (mm).

| - | S0 | S10 | S20 | S30 | S10-M1 | S10-M1.5 | S10-M2 | S10-SF1 | S10-SF2 | S10-SF3 | S60-SF2 | S100-SF2 |
|---|----|-----|-----|-----|--------|----------|--------|---------|---------|---------|---------|----------|
| Fluidity | 230 | 235 | 238 | 245 | 215 | 227 | 232 | 225 | 210 | 200 | 235 | 242 |

#### 3.1.2. Setting Times

It can be seen in Table 9 that the initial setting times of AASSC were between 20 min and 66 min, and the final setting times were between 32 min and 74 min. When the content of silicon manganese slag increased, the setting time of slurries increased, which alleviated the problem of the fast setting time of alkali-activated concrete. The rapid condensation of alkali-activated concrete was due to the internal C-S-H composites accelerated by $[SO_4]^{4-}$ [39].

The reason that the addition of silicomanganese slag increased the setting time of alkali-activated concrete is that silicomanganese slag is less active than GBFS. The setting time of the material decreased with the increase of modulus $M_S$; this is because the activators with higher modulus $M_S$ have higher $[SO_4]^{4-}$ contents and PH values, which can enhance the hydration of slurry [40]. The addition of steel fiber had a small effect on the initial and final setting times of the alkali-activated system; the difference in initial setting time was 7 min, and the difference in final setting times was 5 min between slurries with different fiber contents. The mixture with a higher fiber content had a longer setting time because the steel fibers in the mixture diluted the binders then slowed down the hydration.

**Table 9.** Setting times of samples (min).

| - | S0 | S10 | S20 | S30 | S10-M1 | S10-M1.5 | S10-M2 | S10-SF1 | S10-SF2 | S10-SF3 | S60-SF2 | S100-SF2 |
|---|---|---|---|---|---|---|---|---|---|---|---|---|
| Initial | 20 | 25 | 28 | 35 | 35 | 28 | 24 | 28 | 30 | 35 | 48 | 66 |
| Final | 32 | 38 | 39 | 45 | 42 | 36 | 30 | 34 | 35 | 41 | 55 | 74 |

*3.2. Mechanical Strength of Mixtures*

3.2.1. Effects of the Replacement Ratio of Silicomanganese Slag

Figure 6a shows the influence of the content of silicon manganese slag on the compressive strength of AASSC. The strength of alkali-activated concrete mixed with silicomanganese slag increased rapidly in the early stage and gradually slowed down in the later stage, which is consistent with previous research [25]. The reason can be attributed to the early formation of nano-crystalline or amorphous C-A-S-H gels produced through early activation of slag in the alkaline solution. The additional C-A-S-H gels can be generated through the reaction between the unreacted CaO and the $SiO_2$ from fine amorphous quartz [41]. The 28 day compressive strengths of alkali-activated concrete containing 0, 10%, 20% and 30% substitution ratios of silicomanganese slag were 91.5 MPa, 81.4 MPa, 63.5 MPa and 60.8 MPa, respectively. The development of mixtures at 56 days also exhibited the same tendency. The reason is that the silicon manganese slag is less active than the original cementitious materials [42], and the mixture containing a larger content of silicon manganese slag has a lower development speed, which affects the early and final strength of concrete.

It can be seen in Figure 6b that the development of flexural strength of AASSC was similar to its compressive strength, which developed particularly fast in the early stage and increased slowly in the later stage. The 28 day flexural strengths of mixtures with 0, 10%, 20%, and 30% substitution ratios were 12.9 MPa, 10.5 MPa, 8.1 MPa and 7.7 MPa, respectively. The 56 day flexural strengths of the samples also decreased with the increase in silicomanganese slag content. In summary, The sample S10, with a 10% substitution ratio of silicomanganese slag, had 2.2% increased fluidity in comparison to sample S0 without silicomanganese slag, as well as 1.3%, 4.2% decreased fluidities compared to samples S20 (with 20% substitution ratio) and S30 (30% substitution ratio), respectively; the difference is very small and could be neglected. To the compressive strength at 28 days, sample S10 had an 11% decreased value compared to sample S0, and 28.2% and 33.9% higher values than samples S20 and S30, respectively. To the flexural strength at 28 days, sample S10 had an 18.6% decreased value compared to sample S0, and 29.6% and 36.4% higher values than samples S20 and S30, respectively. Therefore, the substitution ratio of silicomanganese slag hardly affected the fluidity of slurry, but seriously influenced the compressive and flexural strengths, so sample S10 had limited strength induction when compared to S0, and was the best among other mixtures.

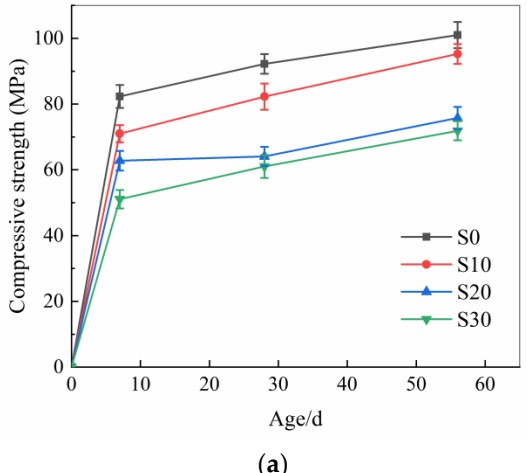 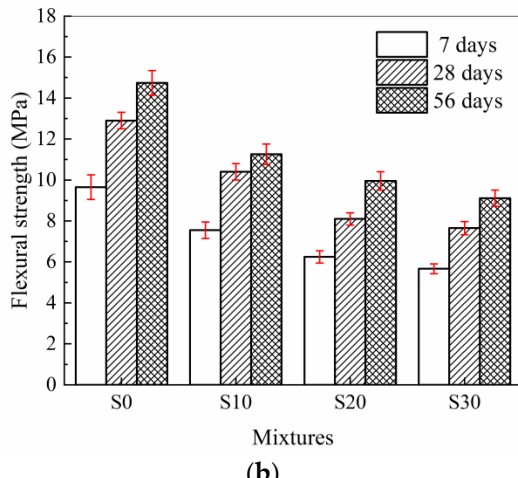

| (a) | (b) |

**Figure 6.** Effects of the content of silicomanganese slag (replacing GBFS) on compressive strength (**a**) and flexural strength (**b**) of AASSC.

### 3.2.2. Effect of Modulus $M_S$

As seen in Figure 7a, the 28 day compressive strengths of AASSC with modulus $M_S$ of 1, 1.5 and 2 reached 71.4 MPa, 73.8 MPa and 79.4 MPa, respectively, and the 56 day compressive strengths of AASSC reached 80.5 MPa, 79.9 MPa and 90.4 MPa, respectively. The compressive strength increased with the modulus $M_S$. The previous study [43] had proven that the higher $M_S$ value can provide more active silicon and generate more silicon–oxygen bonds under the same alkali content, and silicon oxygen bonds are more stable than other bonds, which is conducive to improving their strength. Moreover, water glass has more C-S-H gel than other activating materials in the case of alkali excitation, and C-S-H gel is likely to form a dense structure [44], which indicates that the high modulus can improve the strength of concrete. In general, the strength of the alkali-activated slag system increases with the modulus. However, it had a limit. The modulus of 2.0 is suitable for AASSC in this study.

As shown in Figure 7b, the 28 day flexural strengths of AASSC with 1, 1.5 and 2 modulus reached 9.4 MPa, 10.9 MPa and 11.3 MPa, respectively. When the modulus of water glass was two, the flexural strength of AASSC reached the maximum value. In addition, the flexural strength of AASSC increased with the $M_S$ value, which is consistent with a previous study [45].

### 3.2.3. Effect of Steel Fiber

As seen in Figure 8a, The 28 day compressive strengths of samples with 0, 1%, 2% and 3% steel fiber content reached 79.4 MPa, 85.5 MPa, 99.4 MPa and 116.6 MPa, respectively, which indicates that the compressive strengths of samples with 1%, 2% and 3% fiber contents increased 7.7%, 25.2% and 46.9% than sample without fibers, respectively. The reason is that the steel fibers effectively change the direction of crack propagation in the process of composite cracking, reduce the stress concentration in the crack, and increase the interfacial bonding force in the concrete composite to improve the compressive strength [46].

As seen in Figure 8b, the 28 day flexural strengths of samples with 0, 1%, 2% and 3% steel fiber content reached 11.3 MPa, 12.5 MPa, 14.6 MPa and 16.5 6 MPa, respectively, which indicated that the flexural strengths of samples with 1%, 2% and 3% fiber content increased 10.6%, 29.2% and 46.0% than the sample without fibers, respectively. Furthermore, it was found that the sample without fiber addition had great brittleness, which was characterized by the sudden break when the sample reached its peak load. However, it was different when the concrete incorporated steel fibers, which showed a ductile failure with small cracks in the middle of the AASSC, and the sample did not completely break at the end of the test. Both the compressive and flexural strengths increased with the dosage of steel

fibers [19]. Considering the effects of fluidity, strength and economic factors, the steel fiber content of 2% is more suitable than other samples.

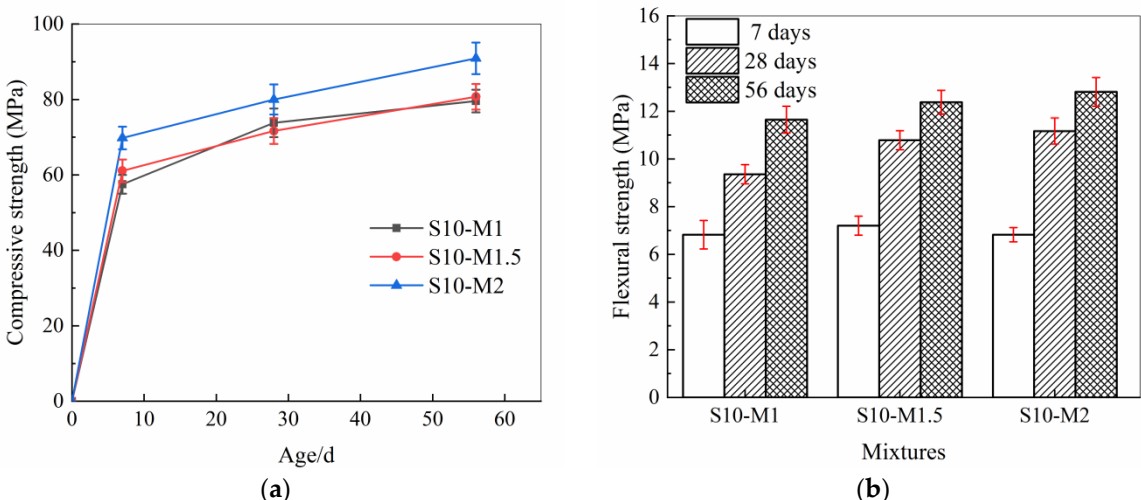

**Figure 7.** Effects of modulus $M_S$ on compressive strength (**a**) and flexural strength (**b**) of AASSC.

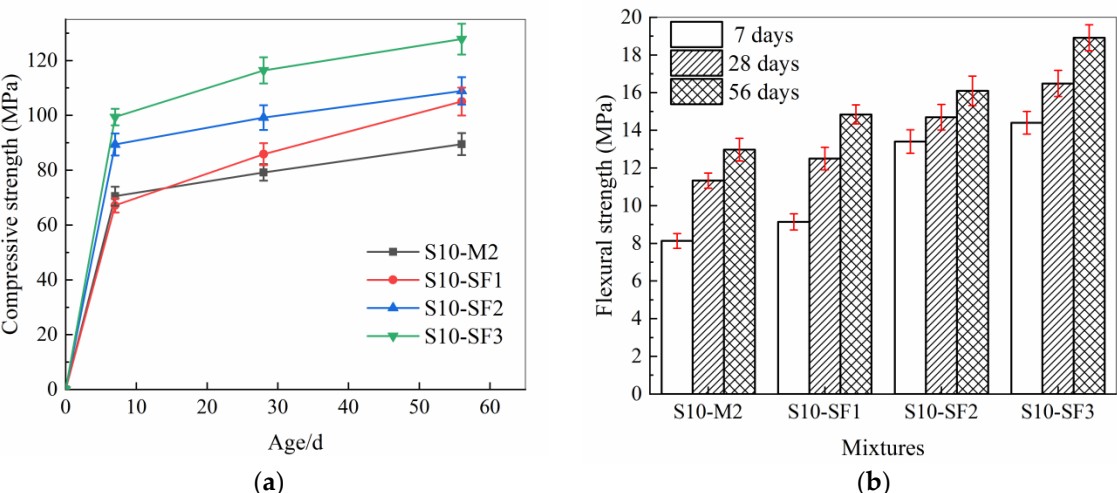

**Figure 8.** Effects of steel fiber content on compressive strength (**a**) and flexural strength (**b**) of AASSC.

### 3.2.4. Massive Substitution Ratio

As seen in Figure 9, with the same content of steel fiber, The 28 day compressive strengths of samples with 60% and 100% substitution ratio of silicomanganese slag to GBFS reached 83.4 MPa and 44.7 MPa, respectively, and the flexural strengths of these samples were 10.1 MPa and 4.1 MPa, respectively. This indicates that considerable silicomanganese slag content is unfavorable to the strength development of AASSC, which also reflects that the excessive silicomanganese slag hardly participates in the alkali-activated polymerization reaction, and eventually affects the mechanical properties of concrete [1]. Therefore, the amount of silicomanganese slag can directly control the final mechanical properties of the AASSC. The test shows that the suitable proportion of silicon manganese slag is 10% when considering high-strength properties and 60% for economic effects.

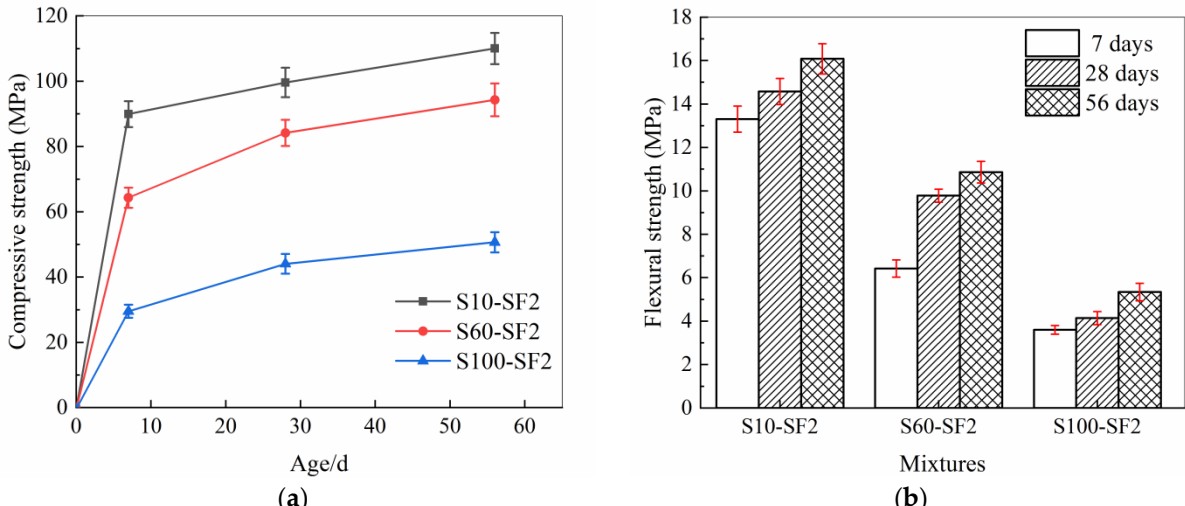

**Figure 9.** Effects of a massive content of silicomanganese slag on compressive strength (**a**) and flexural strength (**b**) of AASSC.

### 3.3. Freeze-Thaw Resistance

#### 3.3.1. Effect of Steel Fiber

As seen in Figure 10, the change rule of compressive strength of GSCC with different fiber contents was the same during the freeze-thaw cycle test. For example, after 100 freeze-thaw cycles, the loss rates of compressive strengths of samples with 1%, 2% and 3% were 25%, 15% and 13%, respectively, which implies that the addition of steel fiber can reduce the loss rate of compressive strength of AASSC. However, the difference in the loss rate of AASSC between 2% and 3% fiber contents was small and could be neglected. The loss rate was the same as the strength loss rate; the mixture with a higher fiber content had less mass lost after the test, mixtures with 2% and 3% both achieved steady improvements compared to other mixtures and there was limited difference between them. Moreover, the mixture S10-M2 without steel fiber addition had more than a 5% mass loss ratio at 700 cycles, which indicates that the sample was broken. The reasons that steel fiber addition decreased the strength and mass loss rate in the test are that the steel fibers restrained the micro-cracks of the matrix under tension [47], which were caused by the freeze-thaw test, and optimized the micro-structure of the matrix [48], so the sample with steel fibers had improved resistance to freeze-thaw cycling. However, a high fiber content could generate flaws in the matrix and induce the resistance to the freeze-thaw test. In summary, 2% steel fiber is suitable for AASSC in the study by considering the strength of the freeze-thaw cycle test and economy.

#### 3.3.2. Effect of Substitution Ratio

As seen in Figure 11, the mass and compressive strength of the samples decreased when the freeze-thaw cycle increased, and there were some large differences between each line, which indicates that a massive substitution ratio has remarkable influence on the concrete resistance to freeze-thaw test. When the cycle reached 300 times, the relative compressive strength ratios of AASSC with the replacement ratios of 10%, 60%, and 100% were 0.84, 0.74, and 0.51, respectively, with 0.4%, 0.9% and 2% mass loss ratios, respectively. Therefore, the AASSC with a higher substitution ratio has larger mass and strength losses. This is because a sample with a higher content of silicomanganese slag tends to have more unreacted components in the matrix [5], which generates a certain amount of holes inside the matrix and make the matrix expand when moisture invades, eventually causing tension and improving the development of micro-cracks. After 300 freeze-thaw cycles, the mass loss rate of AASSC with a 100% substitution ratio of silicon manganese slag reached 6.374%, which indicates the sample was damaged. After 600 freeze-thaw cycles, the mass loss rate of

AASSC with a 60% substitution ratio of silicon manganese slag reached 6.978%; the sample was damaged. In summary, AASSC with 2% steel fiber content and a 10% substitution ratio of silicomanganese slag had the best resistance to the freeze-thaw test, which could still have 75% compressive strength left and only 3.5% mass lost after 800 cycles. With the consideration of massive substitution, AASSC with 2% steel fiber content and a 60% substitution ratio can be applied to construction; it has 3.4% mass lost and 85% strength left after 300 cycles and is destroyed at 600 cycle times.

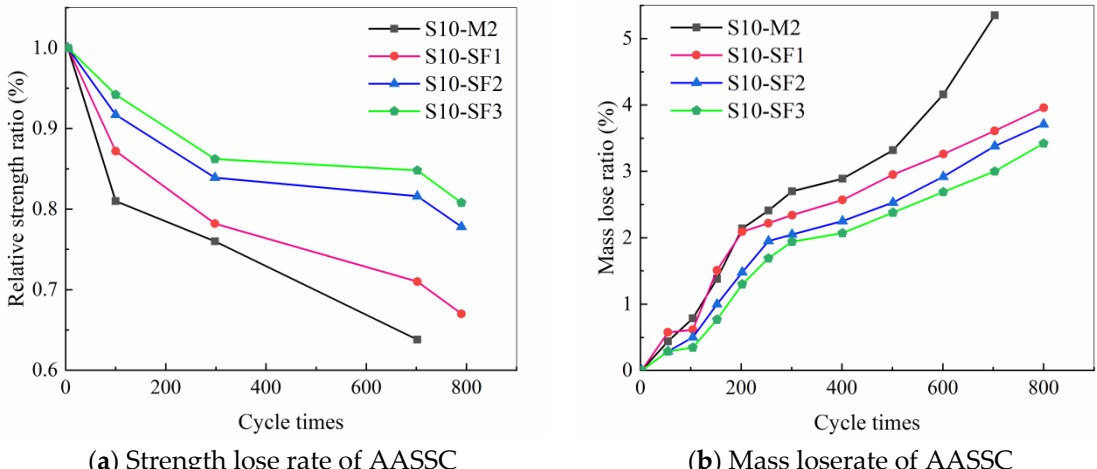

(**a**) Strength lose rate of AASSC          (**b**) Mass loserate of AASSC

**Figure 10.** The influences of different steel fiber content on the freeze-thaw resistance of AASSC where relative strength ratio is that the compressive strength of sample before test versus the compressive strength of sample after test.

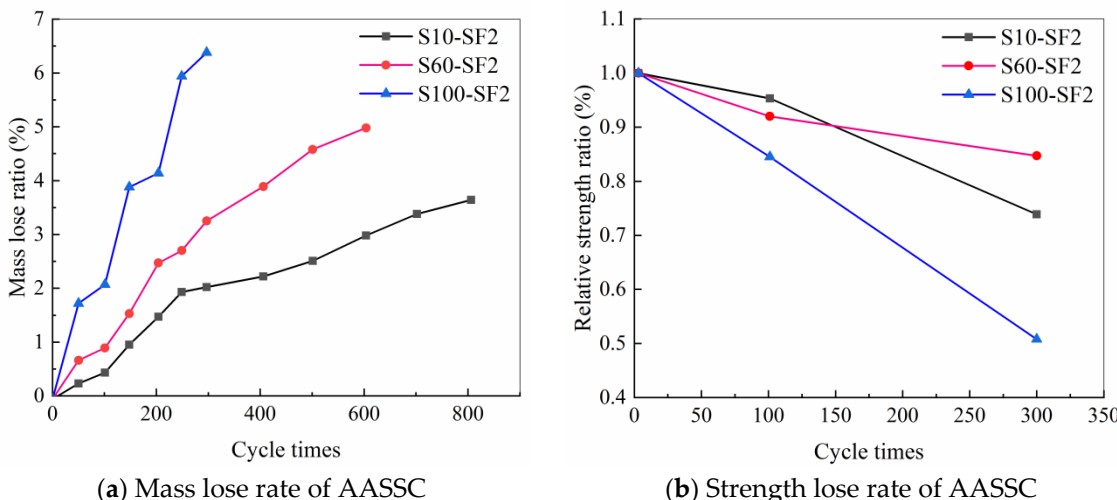

(**a**) Mass lose rate of AASSC          (**b**) Strength lose rate of AASSC

**Figure 11.** The effects of different substitution ratios of silicomanganese slag on sample's resistance to freeze-thaw test.

### 3.4. Scanning Electron Microscope (SEM) Test

3.4.1. Effect of Substitution Ratio on Micro-Structure

It can be seen from in Figure 12(a1–d1) that when the content of silicomanganese slag increased, there were more sporadic holes on the surface of the matrix, which made a looser structure for concrete composite. When the substitution ratio was 30%, the matrix had the largest number of holes with a loose structure on the surface (as shown in (d1), (d2) and (d3)), which implied the compressive strength of that AASSC had a small value. This is consistent with the existing literature [49]. However, the sample S10 with a 10%

substitution ratio had a better surface than S0 without silicomanganese slag. This was maybe because the silicomanganese slag contains hydrated calcium silicate [50] and fine particles that fill the internal gaps of the mortar [6], which improve the adhesion of the particle interfaces. When compared with the micro-structure of S10, S20, and S30, S0 generally had a more compacted micro-structure, which promoted a higher strength in S0. This could explain why compressive and flexural strengths decreased with the increase of the silicomanganese slag content and the phenomenon of freeze-thaw resistance.In addition, as seen in Figure 12(a3–d3), the sample with less silicon manganese slag content had more hydration products. This is because the activity of silicomanganese slag isless than GBFS [49], and mixtures with a higher percentage of GBFS have higher hydration.

The content of silicomanganese slag also directly affected the morphology of hydration products; the hydrated products of the sample S0 were in the form of flakes and knits, crisscrossed and evenly distributed on the surface, and the sample S10 had a small amount of flake hydrated products, but a large amount of knits densely distributed on the surface. The sample S30 had more knits products than S10. This reflected that silicomanganese slag not only affected the hydration process of alkali-activated concrete, but also caused changes in the morphology of hydrated products, which was related to different chemical species, mineralogy and particle sizes, as well as the shape of fly ash and silicomanganese slag, and this ultimately led to differences in reaction and reaction product formation [25]. The silicomanganese slag addition could promote the development of knitted substances in the hydrated products and generate calcium zeolite in the matrix. When compared with the sample S0, there was a denser structure of the net and flocculent gel in the matrix with silicomanganese slag, but the hydration products of S0 had a denser and more even structure in comparison.

### 3.4.2. Effect of Different Modulus on Micro-Structure

The morphology of alkali-silica acid gel generated by the alkali–silica reaction was generally rose petal-type and lamellar-type in Figure 13, which can be presumed to be alkali–silica acid gel by previous research [51–53]. From the comparison of (a1), (b1), and (c1), it can be seen that the collective surfaces of alkali-activated concrete prepared with different modulus $M_s$ were dense and plate-shaped. The interface was more continuous and smoother with the increase of the modulus, and there were few holes in the matrix. It can be seen from the microscopic morphology of the hydrated products in Figure 13(a2–c2) that the hydrated products of AASSC prepared with different modulus were identical and consisted of flocculent gel of hydrated calcium silicate (C-S-H) and some crystals such as calcite, which means that the alkali-activated mortars were hydrated well with applicable strengths and densities. The sample with a higher modulus had more hydrated products and a denser matrix, which is consistent with its strength development. This is because that the activator with a higher modulus has more alkali ($Na_2O$), which plays a vital role during alkali-activation; the system with a higher content of $Na_2O$ had higher pH, which promoted more dissolution of reactive products ($SiO_2$, $Al_2O_3$) [54]. Therefore, the modulus can influence the gel product of the alkali-activated concrete and change the structure and morphology, which affects the flexural and compressive strengths in return, resulting in a sample with a higher modulus that has more uniform and denser hydration products and better compressive and flexural strengths.

As seen from Figure 13(a3–c3), S10–M1 with the modulus 1.0 had many gel substances staggering each other, and some particles were not tightly wrapped; S10–M1.5 with the modulus 1.5 had more sheet products and denser structures than S10–M1; S20–M2 with the modulus 2.0 had quartz and matrix connected, and its dense structure consists of crystals and gels. This indicates that the hydration products in the matrix increase with the modulus and the sample has better-completed hydration with a uniform distribution of hydration products, which eventually improve the mechanical properties of the sample.

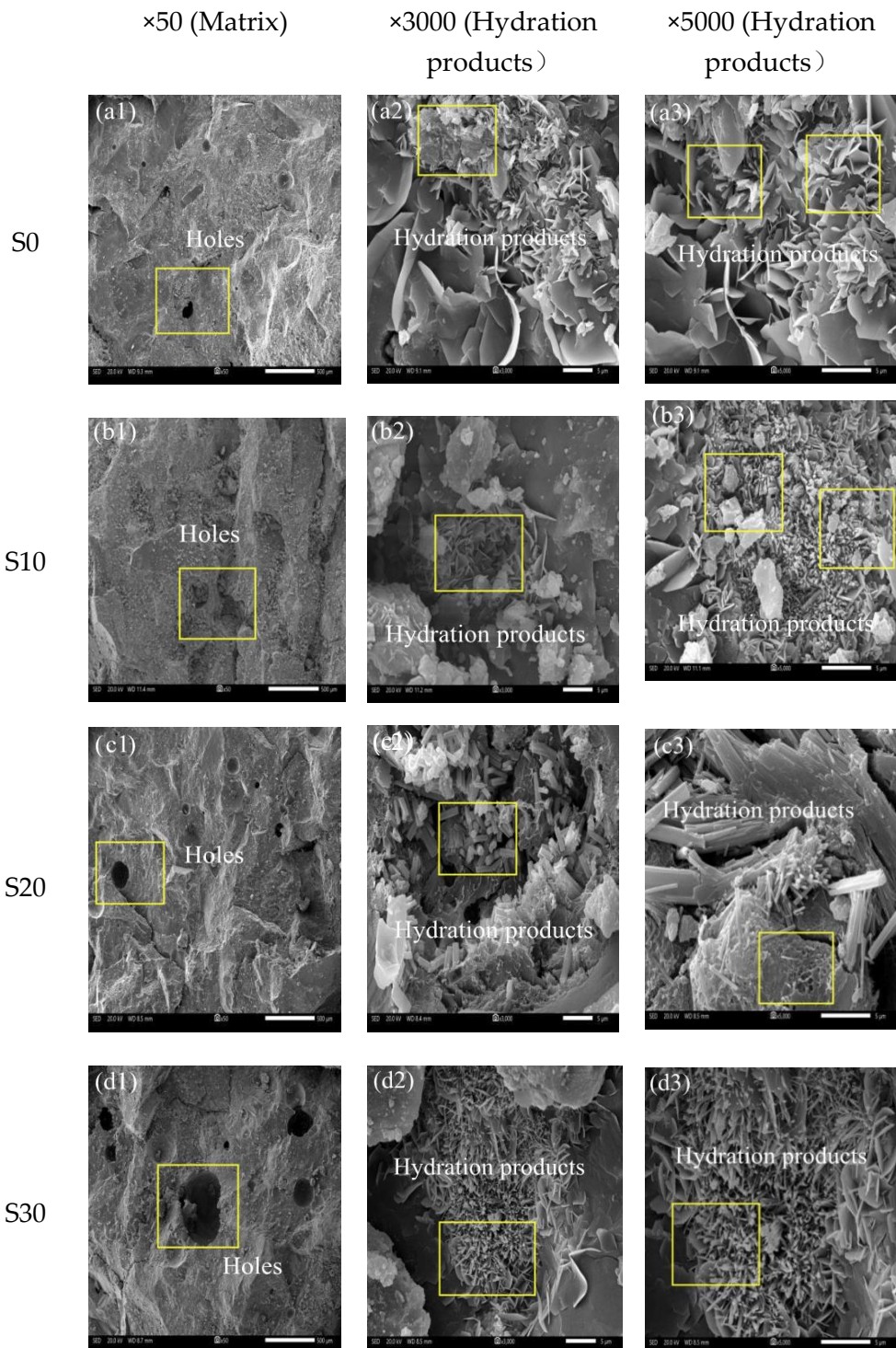

**Figure 12.** SEM images of AASSC with different substitution ratios of silicomanganese slag.

3.4.3. Effect of Steel Fiber on Micro-Structure

It can be seen from in Figure 14(a1–d1) that the bond between steel fibers and the matrix were connected closer when the steel fiber content was higher. When the content of steel fibers reached 3%, there were no obvious micro-cracks and pores at the bonding interface. When the content of steel fibers was 1%, the steel fibers and the matrix were separated, the surrounding holes were obvious with the uneven surface formed the matrix, and the loose connection between steel fibers and the matrix in the interface with obvious debris indicated the poor bonding performance and mechanical strength. This indicates that the

steel fiber content optimizes the bonding performance between steel fibers and the matrix. The reason for this is that the steel fiber is a hydrophilic material; its polar molecules absorb water molecules to form a hydrated product film on the fiber surface, which improves the adhesion between steel fibers and matrix [37]. In summary, the research found that the fibers in the matrix promote the distribution of hydration products to improve the micro-structure of the matrix and enhance the mechanical performance of concrete.

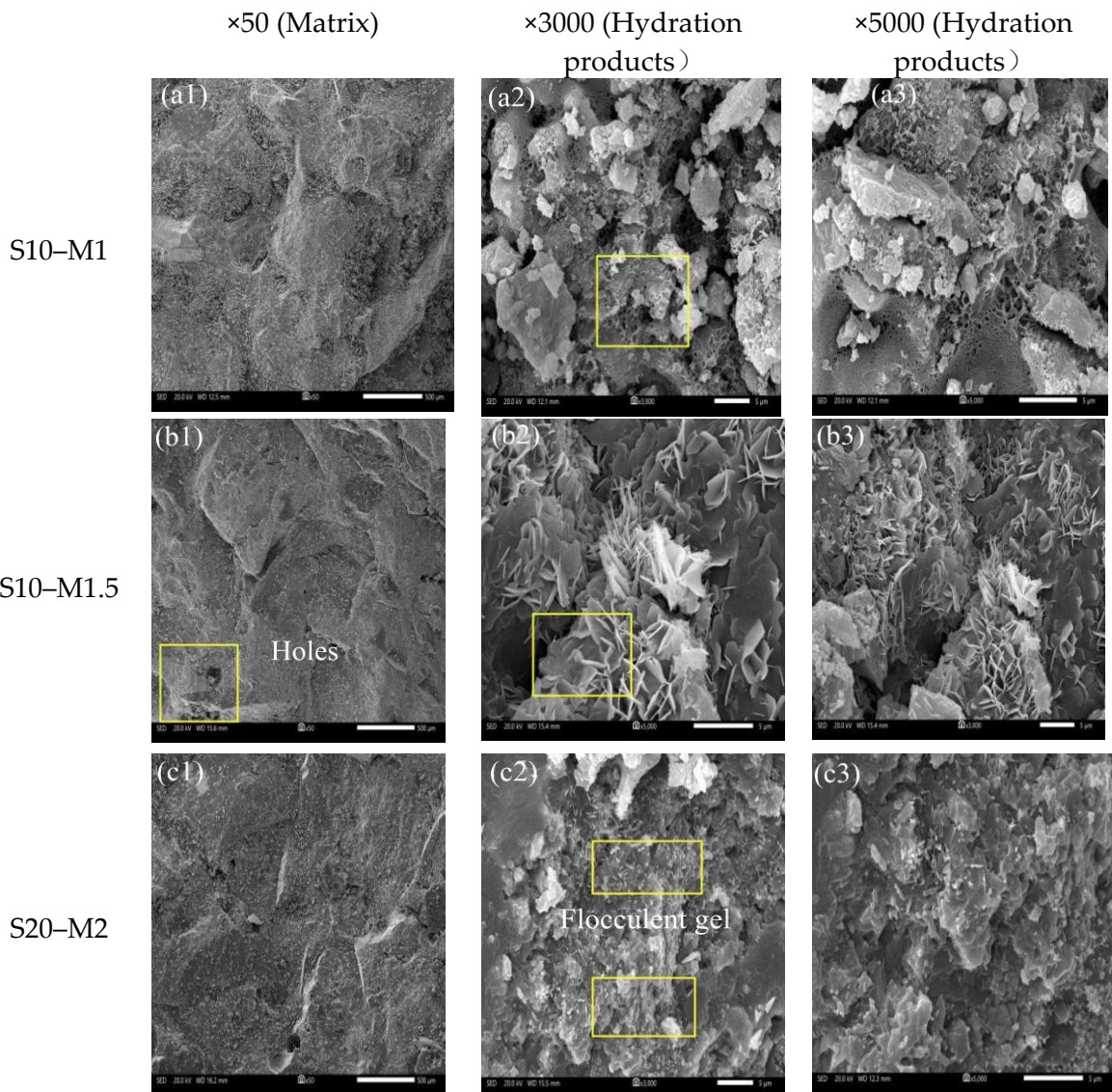

**Figure 13.** SEM images of AASSC with different modulus.

It can be seen from in Figure 14(a2–d2) and Figure 14(a3–d3) that the distribution of hydration products was more and more uniform and became more dense with this sequence; flocculent gel and knitted crystals were vertically and horizontally distributed on the matrix. For example, when compared with the S10 without steel fiber sample, the matrix with the higher steel fiber content had more hydration products with better hydration. Moreover, In the matrix, the steel fibers were tightly wound to form a mesh structure, which improved the bearing capacity of the matrix. The steel fibers traversed and bridged the micro-cracks to bear the tension between micro-cracks, and caused an obvious toughening effect, which effectively improved the mechanical properties of concrete, consistent with a previous study [55]. This explained the phenomenon that AASSC with steel fibers had a higher strength and toughness than the alkali-activated mortar without steel fibers.

×200 (Matrix and Interface between it and fiber)　×3000 (Hydration products）　×5000 (Hydration products）

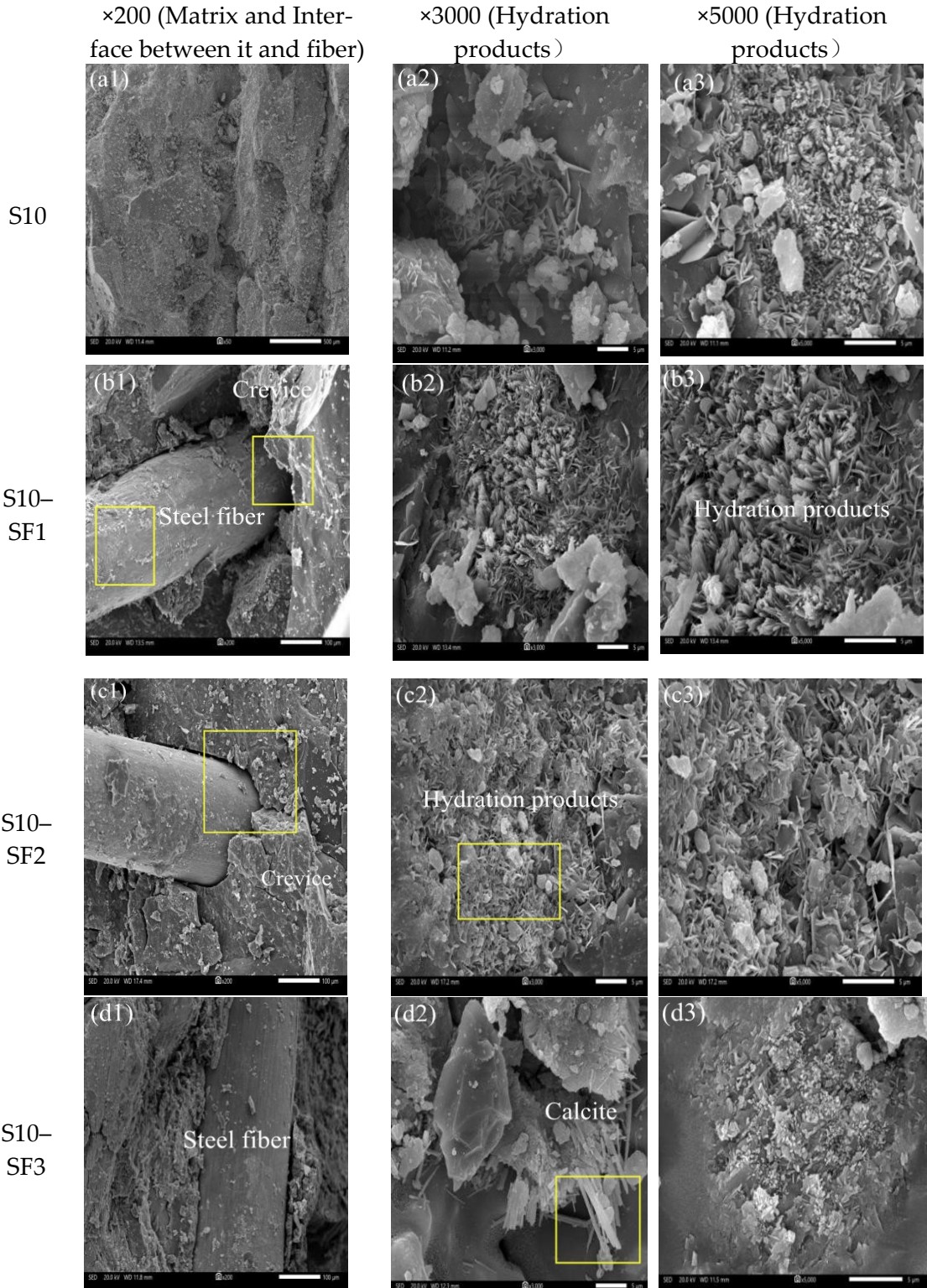

**Figure 14.** SEM images of AASSC with different steel fiber contents.

### 3.4.4. Effect of Massive Silicomanganese Slag Addition on Micro-Structure

As seen in Figure 15(a1–a3), the matrix of S10–SF2 was flat with a dense structure accompanied by considerable matrix debris on the surface of the steel fiber after being pulled out. The steel fibers were closely connected to the matrix on the surface. This indicates that there was a strong adhesive force between steel fibers and the matrix, which had a toughening effect on concrete composite. Moreover, there were many hydrated

products, and the crystals distributed on the matrix, which shows a completed hydration happened in the matrix; as seen in Figure 15(b1–b3), S60-SF2 with 60% silicomanganese slag substitution ratio to GBFS had an irregular matrix surface with few holes and micro-cracks, and the steel fiber was closely connected with the matrix, which indicates that the interface between steel fibers and the matrix was closely bonded. Moreover, there were fewer hydration products than S10–SF2, which indicates S60-SF2 was less hydrated than S60-SF2; as seen in Figure 15(c1–c3), S100-SF2 with 100% silicomanganese slag substitution ratio to GBFS had an uneven matrix with a large number of pores and micro-cracks, which implied the matrix had poor compactness; the steel fiber and the matrix were separated from each other and accompanied by obvious micro-cracks. Moreover, there were few hydration products, with unreacted quartz particles on the surface, which indicated that the matrix had poor mechanical performance and durability. In conclusion, the micro-structure results can vividly express the mechanical properties and durability of AASSC.

### 3.5. XRD Analysis

#### 3.5.1. Effect of Different Contents of Silicomanganese Slag

As shown in Figure 16, the main peaks of the samples from 10 degrees to 80 degrees were quartz ($SiO_2$), calcite ($CaCO_3$) and hydrated calcium silicate (C-S-H), among which the peak of $SiO_2$ was the largest. There was a "convex hull" phenomenon at 25–35° in the figure, which indicates the formation of C-S-H gel generated in the matrix [56]. From the number of diffraction peaks, the number of hydration products and calcites decreased with the increased content of silicomanganese slag. The peak of cobalt chromite ($Ca_2CoSiO_7$) may be caused by silicomanganese slag, which was observed in the XRD test of silicomanganese slag. Calcite may be produced by the reaction of hydrated products with carbon dioxide [57]. As seen in Figure 17, S10 with a 10% content of silicon manganese slag, had a small relative reduction of each main diffraction peak, which indicates that the crystal structure of the material was relatively completed. Therefore, the mechanical properties of the materials prepared with the small content of silicon manganese slag were better than those with a high substitution ratio, which is consistent with the analysis of the micro-structure.

#### 3.5.2. Effect of Different Modulus

It can be seen in Figure 18 that the XRD phases of these three samples were mainly quartz ($SiO_2$), calcite ($CaCO_3$) and hydrated calcium silicate (C-S-H), and there was also a small amount of cobalt chromite. From Figure 19, it can be seen from the diffraction peak that the hydration products and calcite of samples increased with the modulus, but the quartz ($SiO_2$) of each sample had no obvious difference between different modulus. S10-M2 with modulus 2.0 resulted in an increased relative intensity and quantity of each main diffraction peak when compared to other samples, which indicates that the crystal structure of the hydrated product was better completed and explains the reason why S10-M2 had higher strength than other samples. There was no Ca (OH)$_2$ detected in the samples, which may be due to the pozzolanic nature of silica fume and fly ash or the formation of calcite with carbon dioxide.

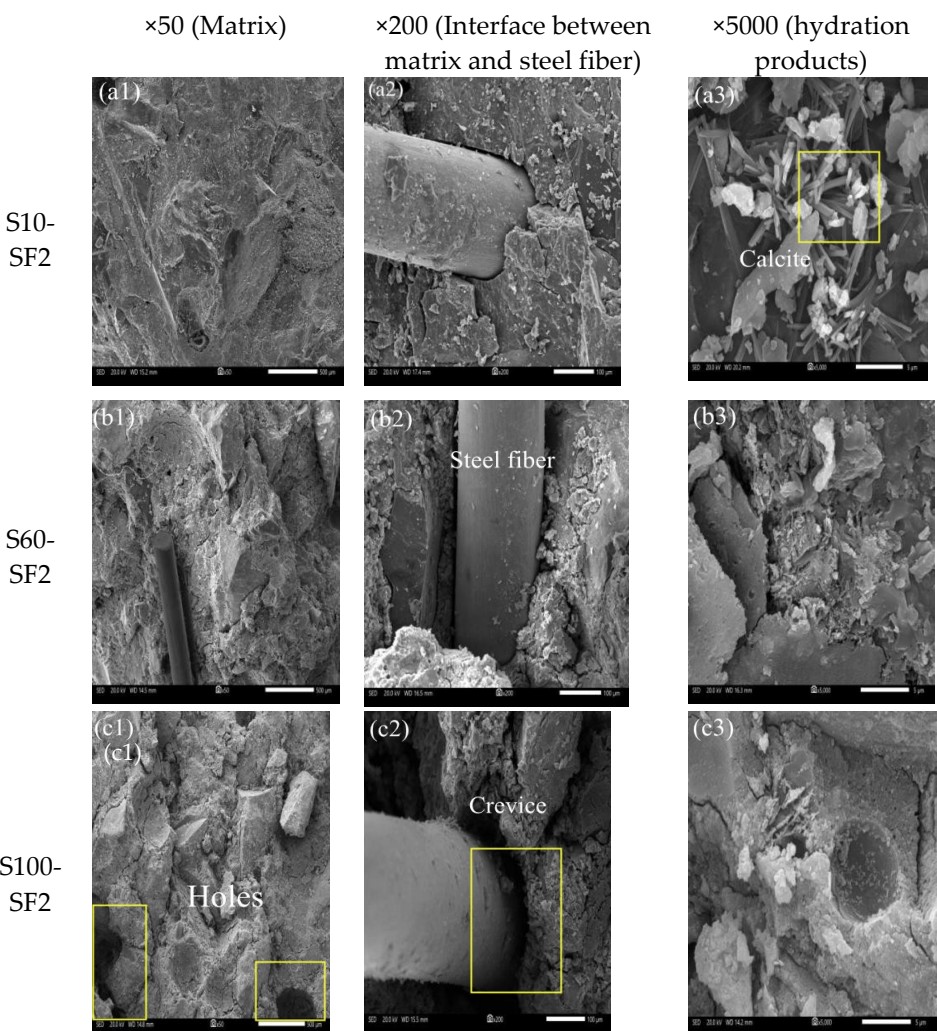

**Figure 15.** SEM images of AASSC with different substitution ratios of silicomanganese slag.

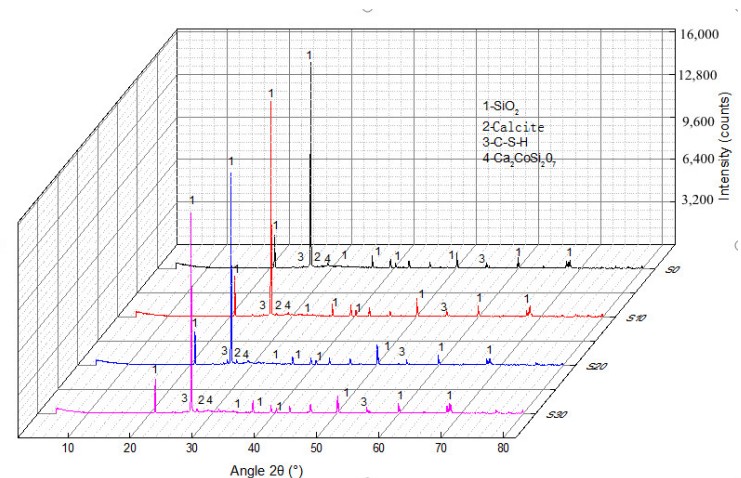

**Figure 16.** XRD patterns of AASSC with different contents of silico-manganese slag.

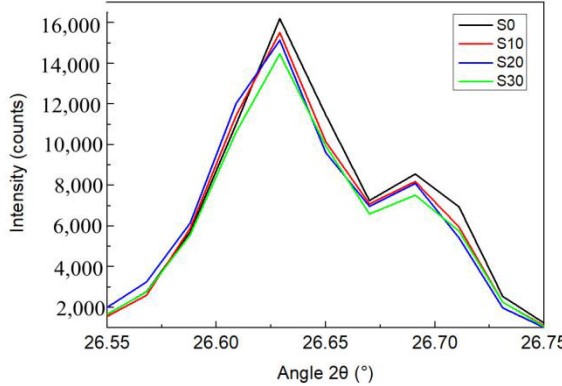

(**a**) SiO$_2$ peak values in different mortars

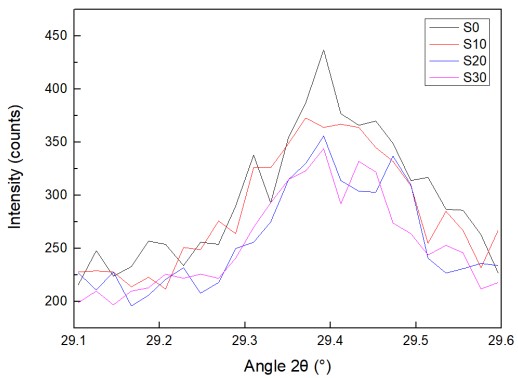

(**b**) C-S-H peak values in different mortars

**Figure 17.** Effect of different contents of silicomanganese slag on the comparison of peak values of SiO$_2$ and C-S-H of samples.

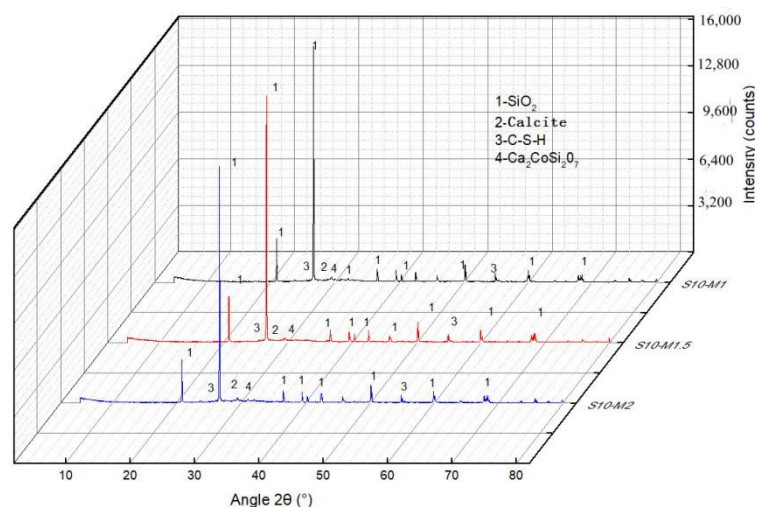

**Figure 18.** XRD patterns of AASSC with different modulus.

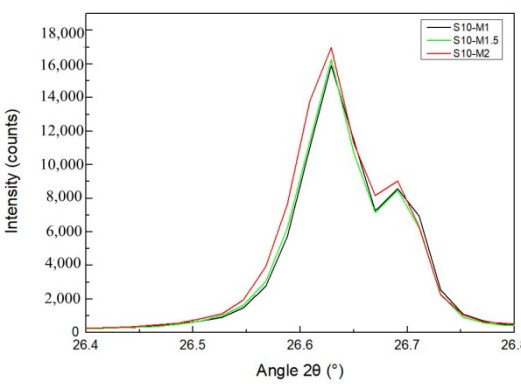

(**a**) SiO$_2$ peak values in different mortars

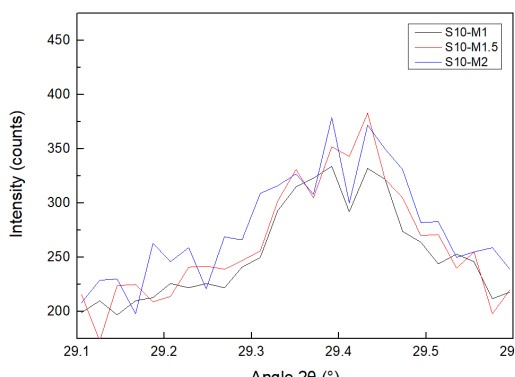

(**b**) C-S-H peak values in different mortars

**Figure 19.** Effect of different modulus on comparison of peak values of SiO$_2$ and C-S-H of samples.

## 4. Conclusions

In this paper, the method of GBFS of alkali-activated concrete partly replaced with silicomanganese slag was studied. The effects of the content of silicomanganese slag, alkali-activated modulus ($M_S$) and steel fiber content on the mechanical properties, freeze-thaw

resistance and micro-structure of alkali-activated silicomanganese slag concrete (AASSC) were deeply investigated. The conclusions can be drawn:

(1) AASSC with a 10% substitution ratio of silicomanganese slag can reach over 80 MPa and over 132 MPa with the 3% steel fiber dosage; it can retain 80% compressive strength left with 3.5% mass lost after 800 freeze-thaw cycles by adding 2% steel fibers, which can allow the AASSC to be successfully applied to practical needs.

(2) ASSC with the higher silicomanganese slag content has lower compressive and flexural strengths, but higher fluidity and setting times. When compared to pure alkali-activated concrete, ASSC with a 10% substitution ratio of silicomanganese slag has a limited strength reduction and good working performance, which is preferable to practical applications.

(3) AASSC with a higher steel fiber content has better mechanical performance and resistance to freeze-thaw cycling. The steel fibers reduce the fluidity and setting times of slurry but can be neglected. The steel fiber content of 2% volume fraction is preferred for real applications in consideration of workability and strength.

(4) The steel fiber addition is important to the freeze-thaw resistance of AASSC, and AASSC with the higher fiber dosage has better resistance. Moreover, despite the harmful effect of large amount of silicomanganese slag on AASSC, AASSC with a 60% substitution ratio of silicomanganese slag was damaged at 600 freeze-thaw cycles and had 3.3% mass lost with 60% strength left; it could also applied to construction considering a massive substitution to GBFS.

## 5. Recommendation

The future research should focus on investigating other properties such as carbonization, projectile strength of AASSC. The application of AASSC on concrete structures is also worth to study in the future. Moreover, preplaced aggregate concrete is a special method that reduces the mortar content with increased economic benefits and is an interest for future AASSC research to solve disposal of recycled waste.

**Author Contributions:** B.L. contributed to the conceptualization, funding acquisition, supervision, and resources of this study; D.W. contributed to the investigation, formal analysis, methodology, writing—original draft of this study; E.M. contributed to the validation, writing—review & editing, visualization of this study. All authors have read and agreed to the published version of the manuscript.

**Funding:** Natural Science Foundation of Hunan Province, China (2020JJ4579).

**Data Availability Statement:** This paper is new. Neither the data nor any part of its content has been published or has been accepted elsewhere.

**Acknowledgments:** This study has been financially supported by the Natural Science Foundation of Hunan Province, China (2020JJ4579).

**Conflicts of Interest:** The authors declared that they have no conflict of interest to this work. We declare that we do not have any commercial or associative interest that represents a conflict of interest in connection with the work submitted.

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
