# Peer review of "Study on Mechanical Properties and Durability of Alkali-Activated Silicomanganese Slag Concrete (AASSC)"

_buildings, doi:10.3390/buildings12101621_

Round 1

Reviewer 1 Report

Paper ID: buildings-1927078

Type:Article
Title:
Study on mechanical properties and durability of geopolymer silicomanganese slag concrete (GSSC)

Authors: Bai fu Luo, Dong Wang, Mohamed Elchalakani

This paper investigates the method of using silicomanganese slag to substitute grounded blast furnace slag in geopolymer.  Although the testing methods and compared results attained in the present study show the importance of the paper, the authors should address the following comments:

  1. Novelty in comparison to recent literature? Need to be emphasized.
  2. The results in the paper have not been adequately discussed by the relevant literature.
  3. Abstract: Geopolymer is the green cementitious material produced by the silica aluminum waste 13 and other substances? Is it correct?
  4. Abstract: abstract required to have a short introduction, problem statement, significant finding, and novelty: Please rewrite.
  5. Does it become a new green concrete when GSSC is used instead of GBFS?
  6. Throughout the text, there are some typos that must be eliminated.
  7. Please revise all subscripts and superscripts throughout the manuscript.
  8. The authors must not lump references together. Rather the contribution from each literature must be stated individually.
  9. In the scientific study, the literature study cannot be given as in the thesis.

10.  Introduction section: You should focus on geopolymer. A clear difference between geopolymer and alkali-activated materials has been established in the literature. The authors should discuss these materials further by considering this distinction.     

  1. There should be a space between number and unit. Please correct these errors in the paper.
  2. How were the mixing ratios chosen?
  3. Why did you choose 90℃ steam curing and 20℃ curing?
  4. The error bar of the figures can help to show the distinction between the samples.
  5. The conclusion part seems to be more like an experimental report rather than a scientific paper. More mechanisms should be added.

Author Response

AUTHORS RESPONSE TO REVIEWER1 COMMENTS

The authors would like to thank the Editor and Reviewers for their thorough review and comments that enhanced the quality of the manuscript.

This paper investigates the method of using silicomanganese slag to substitute grounded blast furnace slag in alkali-activated.  Although the testing methods and compared results attained in the present study show the importance of the paper, the authors should address the following comments:

Before the responses, it should be mentioned that we changed the 'geopolymer silicomanganese slag concrete (GSSC)' into 'alkali-activated silicomanganese slag concrete (AASSC)', because the mortar we utilized should be prefer alkali-activated mortar to geopolymer.

Point 1:Novelty in comparison to recent literature? Need to be emphasized.

Response 1: Thanks for mention it. The novelty  of the method in this study in comparison to silicomanganese slag in cement-based concrete and in alkali-activated concrete were discussed in last paragraph in introduction, display as ' This paper uses silicomanganese slag powder to replace the grounded blast furnace slag to generate alkali-activated silicomanganese slag concrete (AASSC) with excellent performance. The main cementitious material slag or fly ash in alkali-activated concrete generally has more percentage in alkali-activated concrete in comparison to the cement in cement-based concrete, which implies that silicomanganese slag can be utilized higher in alkali-activated concrete than that in cement-based concrete. Compare to silicomanganese slag in alkali-activated concrete, the method in this study can have the grounded blast furnace slag and silicomanganese slag both activated by sodium hydroxide, and silicomanganese slag powder can also act as a filler to fill the micro-voids inside cementitious materials, resulting improved strength and durability'.

Point 2: The results in the paper have not been adequately discussed by the relevant literature.

Response 2: Thanks for mention it. we added more references in the discussion to better illustrate our conclusions.

Point3: Abstract: Alkali-activated is the green cementitious material produced by the silica aluminum waste 13 and other substances? Is it correct?

Response 3: Thanks for mention it. This was not 100% correct and completed, we replaced with 'Alkali-activated materials are produced by chemically polymerizing the aluminosilicates materialsusing alkaline activators, can effectively lower the greenhouse-gas emissions (approximately73%) released by ordinary Portland cement (OPC)', for better express alkali-activated material.

Point4: Abstract: abstract required to have a short introduction, problem statement, significant finding, and novelty: Please rewrite.

Response 4: Thanks for mention it. The short introduction and problem statement were added as 'Alkali-activated  materials are produced by chemically polymerizing the aluminosilicates materials using alkaline activators, can effectively lower the greenhouse-gas emissions (approximately 73%) released by ordinary Portland cement (OPC). Silicomanganese slag is a large solid waste discharged from the ferroalloy industry in China, can pollutes the environment and    occupies resources'. The mechanical strength and freeze-thaw resistance were added with data for represent significant findings, surely, the whole abstract was rewritten.

Point5: Does it become a new green concrete when AASSC is used instead of GBFS?

Response 5: Thanks for mention it. GBFS is a industrial waste, and geoploymer concrete using GBFC could seen as a green concrete. AASSC replacing GBFC with  aluminosilicates materials  in composite could has the potential  solve the dispose of massive aluminosilicates materials, so AASSC is a green material without using GBFC.

Point6: Throughout the text, there are some typos that must be eliminated.

Response 6: Thanks for mention it. we have corrected the typo mistakes.

Point7: Please revise all subscripts and superscripts throughout the manuscript.

Response 7: Thanks for mention it. we have corrected the subscripts and superscripts

Point8: The authors must not lump references together. Rather the contribution from each literature must be stated individually.

Response 8: Thanks for mention it. we recited the references in the manuscript with individuals

Point9: In the scientific study, the literature study cannot be given as in the thesis.

Response 9: Thanks for mention it. we recited the references to make it right.

Point 10: Introduction section: You should focus on alkali-activated. A clear difference between alkali-activated and alkali-activated materials has been established in the literature. The authors should discuss these materials further by considering this distinction.     

Response 10: Thanks for mention it. It is actually alkali-activated concrete in the study, so we deleted the geopolymer and replaced with alkali-activated concrete for clearness.

Point11: There should be a space between number and unit. Please correct these errors in the paper.

Response 11: Thanks for mention it. we corrected the problem with pace between number and unit.

Point12: How were the mixing ratios chosen?

Response 12: Thanks for mention it. We now added the explanation of this in article and displayed as 'The mixing ratios were derived from previous research [1] and adjusted based on our own raw materials, for raw materials from different factory would have different mechanical properties'.

Point13: Why did you choose 90℃ steam curing and 20℃ curing?

Response 12: Thanks for mention it. 90℃ steam curing method specialized in alkali-activated concrete could reach strength development plateau [1] by enhancing the activation between activator and slag or fly ash, which was also applied  to alkali-activated concrete literature [2, 3,4]

Point14: The error bar of the figures can help to show the distinction between the samples.

Response 14: Thanks for mention it, we collected the data again and drew graphics with error bars for every figures. By the way, we did not apply error bar to the figures in freeze-thaw results, because the data on the Figures were ratios and error bar would made such congestion between these lines.

Point15: The conclusion part seems to be more like an experimental report rather than a scientific paper. More mechanisms should be added.

Response 15: Thanks for mention it, we added more important result with data and rewrote the whole conclusion, also more mechanism were added inside.

Reference

  1. Li,N.; Shi,C.; Zhang,Z.; Zhu,D.; Hwang,H.; Zhu, Y. et al., A mixture proportioningmethod for the development of performance-based alkali-activated slag-basedconcrete, Cement Concr. Compos. 2018, 93, 163
  2. Liu, Y.; Zhan, Z.; Shi,C.J. Development of ultra-high performance alkali-activated concrete (UHPGC): Influence of steel fiber on mechanical properties. Cement and Concrete Composites2020, 103670 - 103670.
  3. Ambily, P.S.; Umarani Ravisankar, Dattatreya, J.K. et al., Development of ultrahigh-performance alkali-activated concrete, Mag. Concr. Res. 2013, 66 (2), 82–89.
  4. Wetzel, A.; Middendorf, B.Influence of silica fume on properties of fresh and hardened ultra-high performance concrete based on alkali-activated slag, Cement Concr. Compos. 2019, 100, 53–59.

Reviewer 2 Report

After a comprehensive review of the manuscript "Study on mechanical properties and durability of geopolymer silicomanganese slag concrete (GSSC)", I am recommending it for publication in this special issue of the journal after processing major revisions to the comments attached in the PDF file.

Author Response

AUTHORS RESPONSE TO REVIEWERS2’ COMMENTS

Analysis of mechanical properties and failure mechanism of preplaced coral aggregate concrete (PCAC)

Baifu Luoa, Dong Wanga,, Mohamed Elchalakani3

The authors would like to thank the Editor and Reviewers for their thorough review and comments that enhanced the quality of the manuscript. The paper has been revised to incorporate the comments and suggestions made by the reviewers. 

REVIEWR’S COMMENTS:  REVIEWR2 provide certain amount of suggestion to adjust the quality of this paper, which is really helpful to improve the paper.

Point 1:Provide summary of results in the abstract.

      Before the responses, it should be mentioned that we changed the 'geopolymer silicomanganese slag concrete (GSSC)' into 'alkali-activated silicomanganese slag concrete (AASSC)', because the bender we utilized should be prefer alkali-activated mortar to geopolymer.

Response1: We provided the summary of results and also adjusted the abstract.

Point 2:It focus on the English grammar and typos.

Response2: We rechecked the grammar mistakes and typos and corrected , and also had professor checked before submit again.

Point 3: It should provide a Table of abbreviations in the paper due the massive usage.

Response3: We provided Table of abbreviations in Table 1.

Point 4:Standardization of writing hyphenated phrases throughout the manuscript, whether with or without hyphens. Apply this note to all words that contain a hyphens (such as alkali activated, Freeze thaw, high temperature, high-strength, ultra high, micro-structure, alkali-silica)

Response4: We applied hyphens in the hyphenated phrases  (such as alkali-activated, Freeze-thaw, high- temperature, high-strength, ultra-high, micro-structure, alkali-silica)

Point 5:  Some acronym needs to be define like GSSC, SEM, S0 and so on.

Response5: we checked the acronyms and defined.

Point 6: It should provide recommendation in the paper for future research.

Response6: We provided the section of recommendation after conclusion.

Point 7: There no reference from 2022, try to add more of this to support.

Response7: We added more references from 2022 in the paper, including introduction, results.

Reviewer 3 Report

The article presents the results of an experimental work that explores the effect of partial replacement of grounded blast furnace slag by silicomanganese slag, where three replacement ratios of 10, 20 and 30% were adopted. In addition, the effects of the alkali activator modulus and the effect of steel fibers were also investigated, where three fiber contents of 1, 2 and 3% were considered. The compressive strength and flexural strength were the only investigated mechanical tests, while other workability and durability tests were also considered in addition to microstructural analysis. The following comments must be considered by the authors to better improve the quality of the article.

1-     The abstract of this article is too limited and approximately presents nothing about the conducted work and its outcomes. The abstract should be a stand-alone section that presents a general but brief picture about the study, where the important parameters of the experimental work must be included in addition to the significance of the presented work. The abstract should also present the most important numeral results and comparisons. Therefore, it is highly recommended to briefly add the most important experimental details and the most significant conclusions.

2-     Introduction, lines 28-30: Add a reference number at the end of the sentence “In 2008, the output of silicon dioxide manganese …”.

3-     Introduction lines 40 and 41: What is the c change? And why to use had instead of has in the sentence “Silicomanganese slag had good activity and c change the performance of concrete”?

4-     For a better organization, the literature paragraphs from Nah et al. to Liu et al. should be merged in one or two paragraphs.

5-      The related literature is reviewed in scientific articles to show the recent developments in the field of study and the gaps of knowledge in this field. Based on the gaps of knowledge that should be highlighted by the reviewed literature, the authors should explicitly introduce the differences between the current study and the available literature. The authors did not specify the gaps of knowledge in using manganese slag to replace the GBF slag in alkali-activated materials, which should be clearly clarified in the last paragraph of the introduction section. What are the new additions of the current study? The authors must clearly highlight the significance and novel points of the conducted work in the last paragraph of the introduction section. None of this is shown in the current version of the article.

6-     Section 2.1 does not show or even mention the properties of the used silica sand and sand. For instance, what are the particle size range and bulk density of the silica sand? And what is the specific gravity of sand? The mentioned quantities must be provided together with the sieve analysis of the used sand.

7-     Similarly, Section 2.1 ignores the properties of the used steel fibers. What are the diameter, length and aspect ratio of the used fibers? What is the ultimate strength and source of these fibers? Steel fibers have different configurations and coating applications; what is the configuration of the used fibers? Was they straight, crimped, hooked-end, or hybrid shaped? These parameters determine the activity of steel fibers as crack bridging and stiffness enhancement elements and therefore must be included.

8-     Section 2.3.2: The used cubes were 40 mm side length and the prisms were 40×40×160 mm. These specimens are too small; don’t we have a size effect problem here? Please discuss this point and support your discussion with literature works on geopolymer concrete.

9-     How many specimen replicants were adopted for each test? Must be clarified in section 2.3.

10- Section 2.3.2, line 193: The authors stated that “and finally placed in a standard curing room”. Define the conditions of curing in the standard curing room, and specify based on what standards this room was considered as a standard curing room?

11- In Section 2.3.2, the authors must clarify the setup of the bending test. For instance, was it a three-point or four-point bending test? If it was a four-point one, what are the flexural and shear spans?

12- Line 196, Section 2.5.3 must be corrected to 2.3.3.

13- What is the shape of samples used for the freeze-thaw test? How many replications were adopted?

14- The quality of the compressive strength and flexural strength figures (Figures 6 to 9) is not adequate. These figures must be reproduced considering larger font size for axis titles, axis ticks and legend. The thickness of the results’ lines inside the figures should also be increased. The colors used for the bar charts are also not suitable, try using lighter colors.

15- In section 3.2.1, lines 286-287: The authors stated that “In summary, considering fluidity, compressive strength and flexural strength, GSSC with 10% substitution ratio of silicon manganese slag is more suitable than 287 other contents for practical application.” As this point can be considered among the most important findings of the article, it must be clarified in more detail. Discuss this conclusion in terms of percentage decrease in strength and percentage increase in fluidity of the four mixtures.

16- As for Figures 6 to 9, the quality of Figures 10 and 11 must be improved considering the same recommendations.

17- Several typos in addition to academic writing and linguistic issues were recognized throughout the whole manuscript. The authors must check the manuscript line-by-line to assure that the revised copy will be free of any similar issues.

Author Response

AUTHORS RESPONSE TO REVIEWER3 COMMENTS

The authors would like to thank the Editor and Reviewers for their thorough review and comments that enhanced the quality of the manuscript.

REVIEWR’S COMMENTS:

The article presents the results of an experimental work that explores the effect of partial replacement of grounded blast furnace slag by silicomanganese slag, where three replacement ratios of 10, 20 and 30% were adopted. In addition, the effects of the alkali activator modulus and the effect of steel fibers were also investigated, where three fiber contents of 1, 2 and 3% were considered. The compressive strength and flexural strength were the only investigated mechanical tests, while other workability and durability tests were also considered in addition to microstructural analysis. The following comments must be considered by the authors to better improve the quality of the article.

1- The abstract of this article is too limited and approximately presents nothing about the conducted work and its outcomes. The abstract should be a stand-alone section that presents a general but brief picture about the study, where the important parameters of the experimental work must be included in addition to the significance of the presented work. The abstract should also present the most important numeral results and comparisons. Therefore, it is highly recommended to briefly add the most important experimental details and the most significant conclusions.

Before the responses, it should be mentioned that we changed the 'geopolymer silicomanganese slag concrete (GSSC)' into 'alkali-activated silicomanganese slag concrete (AASSC)', because the bender we utilized should be prefer alkali-activated mortar to geopolymer.

Response 1: Thanks for mention it. The abstract was rewritten by introducing the status of geopolymer, silicomanganese slag, and also the purpose of this paper, then expressed what the experimental brief was. Finally we collected the important results and conclusions such as data of compressive strengths and freeze-thaw resistance of alkali-activated silicomanganese slag concrete (AASSC) with varied substitution ratios of silicomanganese slag, and also we collected the conclusion ' AASSC with a higher substitution ratio of silicomanganese slag had more unreacted components that generated holes in the matrix and caused strength and durability reductions' in the tail'.

2-     Introduction, lines 28-30: Add a reference number at the end of the sentence “In 2008, the output of silicon dioxide manganese …”.

Response 2: Thanks for mention it. We switched the referenced data to global SiMn alloy production and explained how many slag generated for per ton of the production with cited references, for lacking of reference to the original data.

3-     Introduction lines 40 and 41: What is the c change? And why to use had instead of has in the sentence “Silicomanganese slag had good activity and c change the performance of concrete”?

Response 3: Thanks for mention it. It was a mistake, we corrected with 'Silicomanganese slag had good activity that could affect the performance of concrete'.

4-     For a better organization, the literature paragraphs from Nah et al. to Liu et al. should be merged in one or two paragraphs.

Response 4: Thanks for mention it. The paragraphs of were restructured, and those were merged in two paragraphs. One of them was collected as Silicomanganese slag in cement-based concrete, and  another was collected as Silicomanganese slag in alkali-activated concrete.

5-      The related literature is reviewed in scientific articles to show the recent developments in the field of study and the gaps of knowledge in this field. Based on the gaps of knowledge that should be highlighted by the reviewed literature, the authors should explicitly introduce the differences between the current study and the available literature. The authors did not specify the gaps of knowledge in using manganese slag to replace the GBF slag in alkali-activated materials, which should be clearly clarified in the last paragraph of the introduction section. What are the new additions of the current study? The authors must clearly highlight the significance and novel points of the conducted work in the last paragraph of the introduction section. None of this is shown in the current version of the article.

Response 5: Thanks for mention it.  the specified  gaps of knowledge in using manganese slag to replace the GBF slag in geoplomer material were wrote in last paragraph in introduction,  delineate as ' This paper uses silicomanganese slag powder to replace the grounded blast furnace slag to generate alkali-activated silicomanganese slag concrete (AASSC) with excellent performance. The main cementitious material slag or fly ash in alkali-activated concrete generally has more percentage in alkali-activated concrete in comparison to the cement in cement-based concrete, which implies that silicomanganese slag can be utilized higher in alkali-activated concrete than that in cement-based concrete. Compare to silicomanganese slag in alkali-activated concrete, the method in this study can have the grounded blast furnace slag and silicomanganese slag both activated by sodium hydroxide, and silicomanganese slag powder can also act as a filler to fill the micro-voids inside cementitious materials, resulting improved strength and durability.'

6-     Section 2.1 does not show or even mention the properties of the used silica sand and sand. For instance, what are the particle size range and bulk density of the silica sand? And what is the specific gravity of sand? The mentioned quantities must be provided together with the sieve analysis of the used sand.

Response 6: Thanks for mention it. It is important to provide these properties of silica sand and sand. The main properties of silica fume and sands were now added in Tables 3-5 along with the properties of GBFS.

7-     Similarly, Section 2.1 ignores the properties of the used steel fibers. What are the diameter, length and aspect ratio of the used fibers? What is the ultimate strength and source of these fibers? Steel fibers have different configurations and coating applications; what is the configuration of the used fibers? Was they straight, crimped, hooked-end, or hybrid shaped? These parameters determine the activity of steel fibers as crack bridging and stiffness enhancement elements and therefore must be included.

Response 7: Thanks for mention it. It is important to provide these properties of steel fiber.  The shape, length, Roughness, diameter, density tensile strength elongation were now added and shown in Table 6.

8-     Section 2.3.2: The used cubes were 40 mm side length and the prisms were 40×40×160 mm. These specimens are too small; don’t we have a size effect problem here? Please discuss this point and support your discussion with literature works on alkali-activated concrete.

Response 8: Thanks for mention it. Using cubes 40 mm side length and the prisms 40×40×160 mm is because we apply pure paste without coarse aggregates, which is applicable to alkali-activated concrete, in according to previous studies [1, 2]. It is true that size effect would cause different strengths for samples, the purpose of this study is concentrating the using silicomanganese slag to substitute GBFS and analyse the substitution ratios, steel fiber contents and different alkali-activated modulus on the mechanical and durability of concrete composite, so we will study the size effect on the samples in future research.

 9-     How many specimen replicants were adopted for each test? Must be clarified in section 2.3.

Response 9: Thanks for mention it. It three specimens for each mixture under mechanical test and freeze-thaw test, we now specified in the text in section 2.3.

10- Section 2.3.2, line 193: The authors stated that “and finally placed in a standard curing room”. Define the conditions of curing in the standard curing room, and specify based on what standards this room was considered as a standard curing room?

Response 10: Thanks for mention it. It is 'standard curing room at temperature 20 ± 5 degrees with humidity more than 95% based on standard DL/T5144-2001[3]', we now added this into text.

11- In Section 2.3.2, the authors must clarify the setup of the bending test. For instance, was it a three-point or four-point bending test? If it was a four-point one, what are the flexural and shear spans?

Response 11: Thanks for mention it. we were using three-point bending test based on the Chinese standard GB / T 17671-2020 [4].

12- Line 196, Section 2.5.3 must be corrected to 2.3.3.

Response 12: Thanks for mention it. Section 2.5.3 corrected to 2.3.3.

13- What is the shape of samples used for the freeze-thaw test? How many replications were adopted?

Response 13: Thanks for mention it. It was Three 40 × 40 mm specimens from each mixture applied for freeze thaw cycle test. we now added in the text.

14- The quality of the compressive strength and flexural strength figures (Figures 6 to 9) is not adequate. These figures must be reproduced considering larger font size for axis titles, axis ticks and legend. The thickness of the results’ lines inside the figures should also be increased. The colors used for the bar charts are also not suitable, try using lighter colors.

Response 14: Thanks for mention it. we redrew the all the mechanical strength figures with better quality, considering larger font size for axis titles, axis ticks and legend, and also we changed the bar charts with lighter colors added the error bar in the figures.

15- In section 3.2.1, lines 286-287: The authors stated that “In summary, considering fluidity, compressive strength and flexural strength, AASSC with 10% substitution ratio of silicon manganese slag is more suitable than 287 other contents for practical application.” As this point can be considered among the most important findings of the article, it must be clarified in more detail. Discuss this conclusion in terms of percentage decrease in strength and percentage increase in fluidity of the four mixtures.

Response 15: Thanks for mention it. we discussed the percentage decrease in strength and percentage increase in fluidity of the four mixtures, display as : 'The sample S10 with 10% substitution ratio of silicomanganese slag had 2.2% increased fluidity in comparison of sample S0 without silicon manganese slag, 1.3%, 4.2% decreased fluidities compare to samples S20 (with 20% substitution ratio) and S30 (30% substitution ratio), respectively, the difference was really small and could be neglect. To the compressive strength at 28 days, sample S10 had 11% decreased value compare to sample S0, and 28.2% and 33.9% higher values than samples S20 and S30, respectively. To the flexural strength 28 days, sample S10 had 18.6% decreased value compare to S0, 29.6% and 36.4% higher values than S20 and S30, respectively. Therefore, the substitution ratio of silicomanganese slag hardly affected the fluidity of concrete composite, but seriously to the compressive and flexural strengths, so sample S10 had limited strength induction compare to S0 and was the best among other mixtures'.

16- As for Figures 6 to 9, the quality of Figures 10 and 11 must be improved considering the same recommendations.

Response 16: Thanks for mention it. We did improvement on all figures for better illustrating our results

17- Several typos in addition to academic writing and linguistic issues were recognized throughout the whole manuscript. The authors must check the manuscript line-by-line to assure that the revised copy will be free of any similar issues.

Response 17: We checked on our own more than once about the academic writing and linguistic issues, and also  had professor checked before submit.

Reference

  1. Liu, Y.; Zhan, Z.; Shi,J. Development of ultra-high performance alkali-activated concrete (UHPGC): Influence of steel fiber on mechanical properties. Cement and Concrete Composites2020, 103670 - 103670.
  2. Wu, Z.; Shi, C.; Khayat, K.H. Investigation of mechanical properties and shrinkage of ultra-high performance concrete: Influence of steel fiber content and shape, Composites Part B 2019, 174, 107201.
  3. DL/T5144-2001, Code for construction of hydraulic concrete, Ministry of housing and urban rural development: Beijing, Hebei, The People's Republic of China 2016.
  4. GB/T 17671-2020; Inspection method of cement mortar. Ministry of housing and urban rural development: Beijing, Hebei, The People's Republic of China

Round 2

Reviewer 1 Report

All of my queries were properly addressed in the revised manuscript; Therefore, the manuscript can be accepted as it is.

Author Response

(The authors gave the same response as above.)

Reviewer 2 Report

I recommend the revised manuscript for publication.

Author Response

(The authors gave the same response as above.)

Reviewer 3 Report

Every change, addition or modification to the first version of the article based on the recommendations of the reviewers must be highlighted or colored in the revised manuscript. Without doing this, the reviewer cannot easily follow the conducted actions by the authors. Therefore, a highlighted revised manuscript must be submitted as it is usual in academic publishing.

Author Response

Every change, addition or modification to the first version of the article based on the recommendations of the reviewers must be highlighted or colored in the revised manuscript. Without doing this, the reviewer cannot easily follow the conducted actions by the authors. Therefore, a highlighted revised manuscript must be submitted as it is usual in academic publishing.

Response:  

Sorry make such problems for reviewers during reviewing. To be honest, the person who wrote and revised this paper have never published any paper, so do not know highlight the changes in the revision (It is a bit hard to use the Track changes in the paper when the whole text was written, so writer highlighted some important changes with blue background and the most important changes with Track changes). The responses was highlighted with blue background and red font in the manuscript.

Previous response:

AUTHORS RESPONSE TO REVIEWER3 COMMENTS

The authors would like to thank the Editor and Reviewers for their thorough review and comments that enhanced the quality of the manuscript. 

REVIEWR’S COMMENTS:
The article presents the results of an experimental work that explores the effect of partial replacement of grounded blast furnace slag by silicomanganese slag, where three replacement ratios of 10, 20 and 30% were adopted. In addition, the effects of the alkali activator modulus and the effect of steel fibers were also investigated, where three fiber contents of 1, 2 and 3% were considered. The compressive strength and flexural strength were the only investigated mechanical tests, while other workability and durability tests were also considered in addition to microstructural analysis. The following comments must be considered by the authors to better improve the quality of the article.

1- The abstract of this article is too limited and approximately presents nothing about the conducted work and its outcomes. The abstract should be a stand-alone section that presents a general but brief picture about the study, where the important parameters of the experimental work must be included in addition to the significance of the presented work. The abstract should also present the most important numeral results and comparisons. Therefore, it is highly recommended to briefly add the most important experimental details and the most significant conclusions.

Before the responses, it should be mentioned that we changed the 'geopolymer silicomanganese slag concrete (GSSC)' into 'alkali-activated silicomanganese slag concrete (AASSC)', because the bender we utilized should be prefer alkali-activated mortar to geopolymer.
Response 1: Thanks for mention it. The abstract was rewritten by introducing the status of geopolymer, silicomanganese slag, and also the purpose of this paper, then expressed what the experimental brief was. Finally we collected the important results and conclusions such as data of compressive strengths and freeze-thaw resistance of alkali-activated silicomanganese slag concrete (AASSC) with varied substitution ratios of silicomanganese slag, and also we collected the conclusion ' AASSC with a higher substitution ratio of silicomanganese slag had more unreacted components that generated holes in the matrix and caused strength and durability reductions' in the tail'.

2-     Introduction, lines 28-30: Add a reference number at the end of the sentence “In 2008, the output of silicon dioxide manganese …”.
Response 2: Thanks for mention it. We switched the referenced data to global SiMn alloy production and explained how many slag generated for per ton of the production with cited references, for lacking of reference to the original data.

3-     Introduction lines 40 and 41: What is the c change? And why to use had instead of has in the sentence “Silicomanganese slag had good activity and c change the performance of concrete”?
Response 3: Thanks for mention it. It was a mistake, we corrected with 'Silicomanganese slag had good activity that could affect the performance of concrete'.

4-     For a better organization, the literature paragraphs from Nah et al. to Liu et al. should be merged in one or two paragraphs.
Response 4: Thanks for mention it. The paragraphs of were restructured, and those were merged in two paragraphs. One of them was collected as Silicomanganese slag in cement-based concrete, and  another was collected as Silicomanganese slag in alkali-activated concrete.

5-      The related literature is reviewed in scientific articles to show the recent developments in the field of study and the gaps of knowledge in this field. Based on the gaps of knowledge that should be highlighted by the reviewed literature, the authors should explicitly introduce the differences between the current study and the available literature. The authors did not specify the gaps of knowledge in using manganese slag to replace the GBF slag in alkali-activated materials, which should be clearly clarified in the last paragraph of the introduction section. What are the new additions of the current study? The authors must clearly highlight the significance and novel points of the conducted work in the last paragraph of the introduction section. None of this is shown in the current version of the article.
Response 5: Thanks for mention it.  the specified  gaps of knowledge in using manganese slag to replace the GBF slag in geoplomer material were wrote in last paragraph in introduction,  delineate as ' This paper uses silicomanganese slag powder to replace the grounded blast furnace slag to generate alkali-activated silicomanganese slag concrete (AASSC) with excellent performance. The main cementitious material slag or fly ash in alkali-activated concrete generally has more percentage in alkali-activated concrete in comparison to the cement in cement-based concrete, which implies that silicomanganese slag can be utilized higher in alkali-activated concrete than that in cement-based concrete. Compare to silicomanganese slag in alkali-activated concrete, the method in this study can have the grounded blast furnace slag and silicomanganese slag both activated by sodium hydroxide, and silicomanganese slag powder can also act as a filler to fill the micro-voids inside cementitious materials, resulting improved strength and durability'.

6-     Section 2.1 does not show or even mention the properties of the used silica sand and sand. For instance, what are the particle size range and bulk density of the silica sand? And what is the specific gravity of sand? The mentioned quantities must be provided together with the sieve analysis of the used sand.
Response 6: Thanks for mention it. It is important to provide these properties of silica sand and sand. The main properties of silica fume and sands were now added in Tables 3-5 along with the properties of GBFS.

7-     Similarly, Section 2.1 ignores the properties of the used steel fibers. What are the diameter, length and aspect ratio of the used fibers? What is the ultimate strength and source of these fibers? Steel fibers have different configurations and coating applications; what is the configuration of the used fibers? Was they straight, crimped, hooked-end, or hybrid shaped? These parameters determine the activity of steel fibers as crack bridging and stiffness enhancement elements and therefore must be included.
Response 7: Thanks for mention it. It is important to provide these properties of steel fiber.  The shape, length, Roughness, diameter, density tensile strength elongation were now added and shown in Table 6.

8-     Section 2.3.2: The used cubes were 40 mm side length and the prisms were 40×40×160 mm. These specimens are too small; don’t we have a size effect problem here? Please discuss this point and support your discussion with literature works on alkali-activated concrete.
Response 8: Thanks for mention it. Using cubes 40 mm side length and the prisms 40×40×160 mm is because we apply pure paste without coarse aggregates, which is applicable to alkali-activated concrete, in according to previous studies [1, 2]. It is true that size effect would cause different strengths for samples, the purpose of this study is concentrating the using silicomanganese slag to substitute GBFS and analyse the substitution ratios, steel fiber contents and different alkali-activated modulus on the mechanical and durability of concrete composite, so we will study the size effect on the samples in future research.

 9-     How many specimen replicants were adopted for each test? Must be clarified in section 2.3.
Response 9: Thanks for mention it. It three specimens for each mixture under mechanical test and freeze-thaw test, we now specified in the text in section 2.3.

10- Section 2.3.2, line 193: The authors stated that “and finally placed in a standard curing room”. Define the conditions of curing in the standard curing room, and specify based on what standards this room was considered as a standard curing room?
Response 10: Thanks for mention it. It is 'standard curing room at temperature 20 ± 5 degrees with humidity more than 95% based on standard DL/T5144-2001[3]', we now added this into text.

11- In Section 2.3.2, the authors must clarify the setup of the bending test. For instance, was it a three-point or four-point bending test? If it was a four-point one, what are the flexural and shear spans?
Response 11: Thanks for mention it. we were using three-point bending test based on the Chinese standard GB / T 17671-2020 [4].

12- Line 196, Section 2.5.3 must be corrected to 2.3.3.
Response 12: Thanks for mention it. Section 2.5.3 corrected to 2.3.3.
13- What is the shape of samples used for the freeze-thaw test? How many replications were adopted?
Response 13: Thanks for mention it. It was Three 40 × 40 mm specimens from each mixture applied for freeze thaw cycle test. we now added in the text.
14- The quality of the compressive strength and flexural strength figures (Figures 6 to 9) is not adequate. These figures must be reproduced considering larger font size for axis titles, axis ticks and legend. The thickness of the results’ lines inside the figures should also be increased. The colors used for the bar charts are also not suitable, try using lighter colors.
Response 14: Thanks for mention it. we redrew the all the mechanical strength figures with better quality, considering larger font size for axis titles, axis ticks and legend, and also we changed the bar charts with lighter colors added the error bar in the figures.

15- In section 3.2.1, lines 286-287: The authors stated that “In summary, considering fluidity, compressive strength and flexural strength, AASSC with 10% substitution ratio of silicon manganese slag is more suitable than 287 other contents for practical application.” As this point can be considered among the most important findings of the article, it must be clarified in more detail. Discuss this conclusion in terms of percentage decrease in strength and percentage increase in fluidity of the four mixtures.
Response 15: Thanks for mention it. we discussed the percentage decrease in strength and percentage increase in fluidity of the four mixtures, display as : 'The sample S10 with 10% substitution ratio of silicomanganese slag had 2.2% increased fluidity in comparison of sample S0 without silicon manganese slag, 1.3%, 4.2% decreased fluidities compare to samples S20 (with 20% substitution ratio) and S30 (30% substitution ratio), respectively, the difference was really small and could be neglect. To the compressive strength at 28 days, sample S10 had 11% decreased value compare to sample S0, and 28.2% and 33.9% higher values than samples S20 and S30, respectively. To the flexural strength 28 days, sample S10 had 18.6% decreased value compare to S0, 29.6% and 36.4% higher values than S20 and S30, respectively. Therefore, the substitution ratio of silicomanganese slag hardly affected the fluidity of concrete composite, but seriously to the compressive and flexural strengths, so sample S10 had limited strength induction compare to S0 and was the best among other mixtures'.

16- As for Figures 6 to 9, the quality of Figures 10 and 11 must be improved considering the same recommendations.
Response 16: Thanks for mention it. We did improvement on all figures for better illustrating our results.

17- Several typos in addition to academic writing and linguistic issues were recognized throughout the whole manuscript. The authors must check the manuscript line-by-line to assure that the revised copy will be free of any similar issues.
Response 17: We checked on our own more than once about the academic writing and linguistic issues, and also  had professor checked before submit.

Reference

1. Liu, Y.; Zhan, Z.; Shi, C.J. Development of ultra-high performance alkali-activated concrete (UHPGC): Influence of steel fiber on mechanical properties. Cement and Concrete Composites 2020, 103670 - 103670.
2. Wu, Z.; Shi, C.; Khayat, K.H. Investigation of mechanical properties and shrinkage of ultra-high performance concrete: Influence of steel fiber content and shape, Composites Part B 2019, 174, 107201.
3. DL/T5144-2001, Code for construction of hydraulic concrete, Ministry of housing and urban rural development: Beijing, Hebei, The People's Republic of China 2016.
4. GB/T 17671-2020; Inspection method of cement mortar. Ministry of housing and urban rural development: Beijing, Hebei, The People's Republic of China  2020.

Round 3

Reviewer 3 Report

The authors conducted most of the required corrections and modifications.